# The Potamochemical symphony: new progress in the high-frequency acquisition of stream chemical data

Paul Floury[1,2]*, Jérôme Gaillardet[1], Eric Gayer[1], Julien Bouchez[1], Gaëlle Tallec[2], Patrick Ansart[2], Frédéric Koch[3], Caroline Gorge[1], Arnaud Blanchouin[2], and Jean-Louis Roubaty [1]

[1] Institut de Physique du Globe de Paris (IPGP), CNRS and Université Sorbonne Paris-Cité, 1 rue Jussieu, 75238 Paris, France

[2] UR HBAN, Institut national de recherche en sciences et technologies pour l'environnement et l'agriculture, Antony (IRSTEA), France

[3] Endress+Hauser SAS, Huningue, France

Corresponding author. E-mail: floury@ipgp.fr and gaillardet@ipgp.fr

**Abstract.** Our understanding of hydrological and chemical processes at the catchment scale is limited by our capacity to record the full breadth of the information carried by river chemistry, both in terms of sampling frequency and precision. Here, we present a proof-of-concept study of a "lab in the field" called the "River Lab" (RL), based on the idea of permanently installing a suite of laboratory instruments in the field next to a river. Housed in a small shed, this set of instruments performs analyses at a frequency of one every 40 minutes for major dissolved species ($Na^+$, $K^+$, $Mg^{2+}$, $Ca^{2+}$, $Cl^-$, $SO_4^{2-}$, $NO_3^-$) through continuous sampling and filtration of the river water using automated ion chromatographs. The RL was deployed in the Orgeval Critical Zone Observatory, France for over a year of continuous analyses. Results show that the RL is able to capture long-term fine chemical variations with no drift and a precision significantly better than conventionally achieved in the laboratory (up to ± 0.5 % for all major species for over a day and up to 1.7 % over two months). The RL is able to capture the abrupt changes in dissolved species concentrations during a typical 6-day rain event, as

well as daily oscillations during a hydrological low-flow period of summer drought.
Using the measured signals as a benchmark, we numerically assess the effects of a
lower sampling frequency (typical of conventional field sampling campaigns) and of a
lower precision (typically reached in the laboratory) on the hydrochemical signal. The
high-resolution, high-precision measurements made possible by the RL open new
perspectives for understanding critical zone hydro-bio-geochemical cycles. Finally, the
RL also offers a solution for management agencies to monitor water quality in quasi
real-time.
**1 Introduction**
Rivers are messengers from the Critical Zone. The chemical composition of rivers
offers a window into the multiple processes that operate among water, organic matter,
primary and secondary minerals and living organisms at the Earth's surface. (Calmels et
al. 2011; Feng et al., 2004; Kirchner et al., 2000; Kirchner et al., 2001; Neal et al., 2012;
Neal et al. 2013). Understanding the parameters that control the composition of river
water is not only a scientific challenge, but also one of the major challenges for
humanity to access and preserve drinkable water (Bain et al., 2012; Banna et al., 2013;
Bartam and Ballance, 1996). A limit in our understanding of water geochemistry at the
Earth's surface is limited by the temporal resolution at which sampling can be operated
(Whitehead et al., 2009). As summarized by J. Kirchner: "If we want to understand the
full symphony of catchment hydrochemical behaviour, then we need to be able to hear
every note." (Kirchner et al., 2004, page 1358). Yet, taking high-frequency sample sets
back to the laboratory, filtering and analysing them for several elements is limited by
the requirement of considerable human resources (Chapman et al., 1996; Danielsen et
al., 2008; Halliday et al., 2015; Neal et al. 2013; Rozemeijer et al., 2014; Strobl and
Robillard, 2008; Telci et al., 2009).
A significant number of studies have reported high-frequency chemical measurements
in watersheds. Thus far, these data have been mostly acquired during limited periods of
time such as single storm events or a day (Beck et al., 2009; Brick et al., 1996;
Chapman et al., 1997; Gammons et al., 2007; Kurz et al., 2013; Liu et al., 2007; Morel
et al., 2009; Montety et al., 2011; Neal et al., 2002; Nimick et al., 2011; Nimick et al.,
2005; Takagi et al., 2015; Tercier-Weaber et al., 2009). Although these studies clearly
highlighted the wealth of information provided by sampling rivers at sub-hourly
frequency, they underestimate the legacy of past hydrological episodes (Kirchner 2006;
Jasechko et al., 2016; Rode et al., 2016) and are of limited use when mass budgets are to
be calculated for a typical hydrological cycle.
To date, the best combination of high-frequency and long-term monitoring ever
reported for river chemistry is a 7-hourly frequency sampling over 18 months (Neal et
al., 2012). In this study, the authors demonstrate the "act of discovery" permitted by
such sampling scheme, by showing that the high sampling frequency of river
hydrochemistry over sufficiently long time spans reveals patterns related to
hydrological and biological drivers that are imperceptible at lower sampling frequency.
Automated approaches, developed using probes installed directly in the river
(Rozemeijer et al., 2010a; Macintosh et al., 2011; Cassidy and Jordan 2011; Dabakk et
al., 1999; Glasgow et al., 2004; Zhu et al., 2010; Yang et al., 2008) or online
instrumental devices in which continuously pumped water is injected (Rozemeijer et al.,
2010b; Zabiegala et al., 2010; Jordan and Cassidy 2011) are alternatives to sampling
methods requiring human intervention. Several papers have been published over the last
decade reporting existing devices mostly focused on monitoring dissolved N or P and

organic matter (Clough et al., 2007; Kunz et al., 2012; Aubert et al., 2013a; Aubert et al., 2013b, Escoffier et al., 2016). A recent overview of the potential of available conductivity, dissolved oxygen and carbon dioxide, nutrients, dissolved organic matter, chrlorophyll and Co in situ probes is given by Rode et al. (2016).

A new solution for high-frequency measurement of river chemistry is offered by bringing the laboratory's measuring devices to the field (the "lab in the field" concept). A Swiss group has recently developed such a system (von Freyberg et al., 2017) by installing ionic chromatography devices in a hut next to a stream. In this paper, we present a parallel initiative named the River Lab (RL) and funded by the French program CRITEX: "Innovative sensors for the temporal and spatial EXploration of the CRITical Zone at the catchment scale" (https://www.critex.fr). This approach, like the previously published one, overcomes traditional limitations on the number of samples and avoids several issues related to sample transport, filtration and storage. The RL is able to perform a complete chemical analysis of all inorganic major anionic and cationic species in the dissolved load of river water using ion chromatography (IC), with a frequency of up to one complete measurement every 40 minutes.

This article is a proof-of-concept paper that describes the analytical design of the RL and its performance by evaluating the precision, reproducibility and accuracy of concentration measurements. The first results from the RL reveal a significant improvement in reproducibility compared to conventional sampling and analysis techniques. Leveraging these optimal analytical conditions, the RL is able to reveal temporal patterns of river chemistry, such as daily concentration variations. The RL opens thus new opportunities in the field of river chemistry research and environmental monitoring.

## 2 Monitoring site

The RL was installed in the Orgeval, Critical Zone Observatory located 70 km eastward from Paris, France (https://gisoracle.irstea.fr/), a temperate agricultural catchment, within the Seine river watershed, and part of the French Critical Zone Research Infrastructure OZCAR ("Observatoires de la Zone Critique, Applications et Recherche"). Orgeval catchment is one of the most instrumented and documented river observatories in France, with 50 years of hydrological data (Garnier et al., 2014). Catchment hydrologic data are available on the ORACLE website (https://bdoh.irstea.fr/ORACLE/).

The RL is installed at the outlet of the Avenelles River, a sub-catchment in the Orgeval watershed. The Avenelles River drains an area of 45 km$^2$. The climate is temperate and oceanic, with cool winters (mean temperature 3°C), warm summers (20°C in average) and an annual precipitation rate of ~650 mm on average. The Avenelles sub-catchment sits within the sedimentary carbonate-dominated Paris Basin. The river is perennial, supplied by groundwater from the Brie aquifer; with water chemistry dominated by $Ca^{2+}$, $SO_4^{2-}$, $HCO_3^{2-}$ and $NO_3^-$ ions. The water level at the Avenelles gauging station shows an average daily volumetric flow rate of 0.2 m$^3$/s (from 1962 to 2016) with low water period in summer (0.1 m$^3$/s) and flash flood events reaching 10.4 m$^3$/s in spring.

## 3 Design of the River Lab

The concept of the RL is to pump river water and feed it to a set of physico-chemical probes and ion chromatography instruments (IC) for a complete analysis of major dissolved species continuously at high frequency (40 minutes is needed for a complete analysis). All the instruments of the RL fit into an isolated bungalow of 4-m length by 2.5-m width, kept at 24°C ± 2°C. The RL was designed by IPGP (Institut de Physique

du Globe de Paris, France) and IRSTEA  (Institut national de Recherche en Sciences et
Technologies pour l'Environnement et l'Agriculture, France) and assembled by Endress
& Hauser (E+H[®]) (Fig. 1). A technical sketch is available in supplementary information
(Fig. SI1).

The RL has been designed around a primary circuit, which pumps the river water at 700
liters per hour. First, the unfiltered river water sampled in the middle of the stream (Fig.
1) continuously supplies an overflow tank where 6 parameters are measured: pH,
conductivity, dissolved $O_2$, dissolved organic carbon (DOC), turbidity and temperature.
The water is then released into the river downstream from the RL. The turnover time of
water in this primary circuit is 2 minutes. The turbidity probe is installed upstream of
the overflow tank in a pipe perpendicular to the flow to provide more accurate
measurements. The turbidity and DOC probes benefit from an automatic self-cleaning
every 5 minutes using compressed air. For all probes, the frequency of acquisition is
one measurement per minute. The tank and each probe are hand-cleaned weekly. All
probes are developed and provided by Endress & Hauser (E+H[®]).

Second, a fraction of water pumped through the primary circuit feeds another circuit
directed toward two IC instruments for the measurement of major dissolved species
concentrations. A filtration system is deployed between the primary circuit and the IC
instruments, consisting of a tangential filter with a 2-μm pore size, followed by a 0.2-
μm frontal filtration system through cellulose acetate filters (Fig. 1) crucial for the IC
instruments. Cation and anion chromatographs, connected in series, are fed
simultaneously every 40 minutes from the filtered water circuit through a injection
valve. Between two injections, the water in the filtered circuit is constantly renewed (1
L per hour). Our tests show that the frequency for a complete and uncontaminated
analyse of cation and anion is actually limited by the filtration device (see part 4.3).
The IC analysis is performed using two Dionex$^®$ ICS-2100 (Thermo Fisher Scientific$^®$)
instruments using eluent produced with concentrated eluent cartridges and ultra-pure
water (Fig. 1). The cationic species measured are $Na^+$, $K^+$, $Mg^{2+}$ and $Ca^{2+}$, and anionic
species are $Cl^-$, $NO_3^-$ and $SO_4^{2-}$. The chosen analysis time is 30 minutes (40 minutes if
$Sr^{2+}$ concentration measurements are included; see details in SI "Ion Chromatographs
characteristics"). The multiport valve installed upstream of the ICs allows us to check
the drift of the instruments and the background signal by regular introduction of
calibration solutions and pure distilled water (see section 4). Pure distilled water is
regularly (every two weeks) introduced to check the residual noise. Both cationic and
anionic chromatographs are calibrated every two months using synthetic solutions
mimicking the river chemistry, made from 1000-ppm mono-elemental standard
solutions. Two sets of calibration solutions are prepared, one for anions and the second
for cations. The first solution (called "River x1") is prepared based on concentrations of
the river water during summer, i.e. with the highest measured concentrations for most
species. In the second solution, these concentrations are doubled (called "River x2").
Further solutions are produced out of River x1 and x2 through dilution by up to ten-fold
to achieve lower concentrations ("River x0.5; x0.25; x0.1"). The resulting five
calibration solutions cover the entire range of possible natural variability of each species
observed for the Orgeval River, including flood events.

Data from probes and ICs are collected, merged and updated in a single database in real
time. Data from the gauging station (flow discharge and precipitation level) are
automatically added to the database. Several parameters of the RL can be remotely
monitored such as pump activity, pressure, flow and temperature in the primary circuit;
activation of the tangential filtration cleaning system, instrument connection, and
temperature in the bungalow. A set of alarms and sensors controls each key point of the
system. An email is automatically sent in case of dysfunction. Under normal operating
conditions, the RL needs human intervention only once per week.

**4 Analytical performances of the River Lab**
RL data acquisition started on the 12th of June 2015. The reliability of the system was
assessed through 5 different tests involving IC measurements and the sampling
procedure (accuracy, drift, precision of the whole system, cross-contamination and
reproducibility). We refer to the 3rd edition of JCGM 200-2012 (JCGM 2012) for the
terminology used in assessing the performance criteria.

**4.1 Accuracy and instrumental drift**
The aim of the RL is to achieve very high-frequency measurements of river chemistry
over long periods of time (pluriannual). To compensate for any long-term drift in the IC
calibration, instruments are calibrated with a new set of solutions every two months or
after each maintenance operation on the IC instruments. However, calibration drift can
occur over timescales shorter than two months, resulting in systematic and / or random
errors in concentration measurements. We evaluated this effect using a set of injections
of the "River x1" solutions, over one week and over two months, (Tab. 1). For all
species measured, no systematic variation was observed in the measured concentration
of the solution "River x1", showing that at the two timescales, instrumental drift does
not induce any systematic bias on concentration measurements, and that most of the
error is of random nature. Therefore, the standard deviation of the concentration
measurements of a given solution can be used as a reliable measure of the error due to
instrumental drift.  The measurement error over one week is calculated as the standard
deviation of concentration measurements over 19 injections of solution "River x1"
performed every 8 hours during one week (from the 5[th] to the 12[th] of November 2015).
The measurement error over two months is calculated as the standard deviation of
concentration measurements over a series of injections performed every two days
during two months (from the 28[th] December 2015 to the 26[th] February 2016). These
error estimates are lower than 1 % over one week and lower than 1.7 % over two
months (Tab.1). The agreement between the calculated concentrations of the "River x1"
solution and the RL measurements also demonstrate the accuracy of the prototype (Tab.

211  1).


**4.2 Precision of the whole system**
In order to estimate the precision of the whole system (IC instruments combined with
the sampling device including the primary circuit, the pump and the filtration units), we
performed a "closed-loop experiment" over the course of one day by connecting the
inlet and the outlet of the primary circuit to a 300-L tank containing river water. The test
was performed three times over two different seasons (the 20[th] of July 2015, the 28[th] of
August 2015, and the 17[th] of April 2016). The conductivity probe (one measurement
every minute) was used to check the stability of the water chemistry during the course
of the experiment (Fig. SI 2). Our results show that a lapse of 2 hours at least is
necessary for the system to stabilize, corresponding to the homogenization time of the
water within the closed loop (Fig. 2). After two hours, major anion and cation
concentrations show a remarkable stability indicating the absence of drift over of 24-
hour time lapse despite the temperature variations in the river water, and allowing us to
estimate the precision of the whole system over one day using the standard deviation of
the measurements performed during the test. The results of the test are presented in
Table 2. The precision reached is lower than 0.5% for all species except for potassium,
for which it is lower than 1.2%.

**4.3 Cross-contamination**
The ability of the RL to detect rapid variations in river chemistry (typically expected
during storm events) depends on 1) the response time of the RL to a perturbation in the
river and 2) the potential cross contamination from one sample to the next one. We
assessed these two effects by a tracer injection experiment. After establishing a closed-
loop experiment (on the $29^{th}$ of August 2015) and allowing for the period of
stabilization, we introduced a known amount of NaCl (200 g previously dissolved in a
small amount of river water) into the 300-L tank of river water in order to simulate a
"spike" in the river chemistry. The monitoring of conductivity in the primary circuit
allowed us to follow the propagation of the spike injection into the primary circuit while
$Cl^-$ concentrations measured by the IC every 40 minutes allowed us to follow its
propagation through the filtration devices and IC instruments (Fig. 3). The conductivity
probe shows that the salinity spike is detected very quickly and stabilized after 5
minutes. This indicates that the water in the primary circuit is quickly homogenized (in
agreement with the high flow rate of the primary circuit: 700 l/h). Conversely, the $Cl^-$
and $Na^+$ concentrations only reach the expected concentration at the second IC
measurement i.e. after 80 minutes.

The first IC measurement following the spike injection indicates that only 93% of the
final steady-state concentration is reached, revealing a contamination of the $(n)^{th}$ sample
by 7% of the $(n-1)^{th}$ sample. In practice, such a contamination will only be significant if
the instantaneous derivative of river concentration with time is important. In the case of
the Orgeval River, where the RL is deployed, the relative derivative of the concentration
with respect to time is lower than 1% per hour for 90% of the time for all species. In
this case, the cross-contamination induces an error of 0.07% compared to the true
concentration, which means that the effect of cross contamination is negligible
compared to the precision of the RL (see section 4.2). However, in the case of flood
events, when the stream flow increases quickly, the derivative of concentration can
change by more than 10% per hour. In such cases, cross contamination will induce an
error of 1% or more. The injection test shows that the time resolution of the RL is
limited by the transfer time of the water between sampling and injection into the IC
instruments. This transfer time of the water in the RL is mainly due to the design of the
filtration system, which may be improved in the future.

**4.4 Reproducibility: RL *vs* Laboratory**
As a final test for assessing the ability of the RL to record fine natural variations of river
chemistry in comparison to conventional techniques of filtration and analyses in the
laboratory, we focused on two days in the summer of 2015 following long periods
without rain ($21^{st}$ of July 2015 for cations and $19^{th}$ of April 2016 for anions) which
showed very high resolution diurnal variations (<5% relative) in chemical composition
of the Orgeval river. In addition to the analyses made by the RL every 40 minutes, we
conducted hourly sampling of the river by collecting 5 litres of water and filtering it
immediately using a Teflon® frontal filtration unit (Sartorius®) with 0.2-µm porosity
polysulfonether filters. Bottles of acidified (at pH = 2) and unacidified river water were
transported to the laboratory at IPGP for measurement of major cations and anions,
respectively, using IC devices similar to those installed in the RL (Thermo Fisher[®] ics
2100). In the laboratory, measurements were performed using Thermo Fisher[®] ics 5000
for cations measurements and Dionex[®] 120 from Thermo Fisher[®] for anions
measurements. The calibration procedure in both laboratory and RL is the same using
the same set of calibration solutions. The error measurement reached in the laboratory is
estimated at 1% through repeated injections of the standard solution "River x1" (every 5
samples). Comparison between the RL and the laboratory for the seven measured
species are shown in Figure 4. First, the measurements made by the RL are more precise
than those performed in the laboratory, a feature that can be primarily attributed to the
greater stability of the continuously working injection system of the RL. Second, the
fine variations measured by the RL are reproduced in the laboratory, validating the
observed diurnal variations and supporting the reliability of the RL to detect changes on
the order of a percent within a day. The third observation is that small yet systematic
offsets between the two sets of data exist, up to 3% for Mg. One possible explanation
for this difference is that the filtration procedures differed between the RL and the
manual sampling, which may have led to a discrepancy in the concentration
measurements related to the potential for some elements to be hosted in the colloidal
phase (Dupré et al., 1999). In addition, the most accurate measurements were obtained
with the RL rather than with the laboratory equipment because the RL is continuously
processing solutions with a similar matrix, thereby minimizing memory effects and
cross-contamination that can compromise measurements if widely differing samples are
run successively on the same instrument. These features of the measurement protocol,
representative of most laboratory workflows for hydrochemical measurements, are
likely to lead to inaccuracies. Regardless of the observed discrepancy between the two
sets of measurements, we note that variations in concentration recorded by the RL and
measured at the IPGP laboratory have the same amplitudes and are synchronous.

**5. Discussion**
**5.1 What are the benefits of bringing the lab into the field?**
The RL presented above allows us to record continuously, at a high frequency and over
long spans of time, the concentration of 7 major dissolved species in a river system.
Although this is beyond the scope of the present paper , the RL presented here opens
new possibilities for the exploration of the fine structure of hydrochemical evolution at
the catchment scale and for improved understanding of the associated hydrological,
geochemical, and biological processes. From a technical point of view, our study shows
that deploying the conventional laboratory measurement techniques in the field adds
significant value. The tests performed and reported above clearly demonstrate an
improvement in precision compared to the analysis of bottled samples taken back to the
lab. We see three main reasons for this improvement.
1) In a given river, dissolved concentrations typically vary by less than one order of
magnitude when water discharge changes by several orders of magnitude (Godsey et al.,
2009). This constancy allows us to select a relatively narrow range of concentration for
establishing specific calibration curves of the IC instruments, a condition which is rarely
possible in the laboratory where different kinds of samples are analyzed.
2) While in the laboratory samples are injected discretely, in the RL river water samples
are injected as a continuous flow. Thus, the primary circuit and the filtration system
operate continuously at a constant pressure, which supports stable and accurate
analyses.
3) The third factor is the experimental conditions in the bungalow. The temperature is
maintained at 24°C ± 2° (in addition to the 40°C thermostatically-controlled
temperature in the column, precolumn and detection device of the ICs) allowing for
better stability of the IC measurements. Moreover, the RL IC instruments are never
stopped, which favours stability.

**5.2 What is revealed by a higher sampling frequency?**
To our knowledge, the high frequency of measurements (one measurement every 40
minutes) reached by the RL installed on the Orgeval River is the highest ever reported
for stream chemistry over several months. To highlight the corresponding improvement
in the recorded concentration signal, we tested the effect of sampling frequency on the
concentration signal. First, we artificially sub-sampled the RL original signal at two
lower sampling frequencies: every 7 hours (starting October 5[th], 2015 at 10 pm) and
every 24 h. The 7-hourly frequency was chosen to reproduce the sampling frequency of
Neal et al., (2012) made in the Plynlimon watershed, Wales. The daily sampling
frequency is typically what is achievable on the long term by "human grab-sampling" in
the field. Second, we calculated the probability density function (PDF) of concentration
measurements over a given time interval. The use of PDFs allows us to explore the
structure of concentration signals beyond the mean concentration, which constitutes an
important metric for river solute budget, but lacks any insight into the variations in
concentrations that can be used to retrieve information on catchment processes. We
describe the PDF by 3 statistical parameters: mean, standard deviation and skweness.
Skewness indicates the distribution asymmetry, both in magnitude and direction (a
positive skewness means that most values are higher than the mean).  Altogether, the
three parameters account, at first-order, for the structure of a concentration signal. We
compared these three parameters for the computed PDFs to quantify the signal
degradation induced by artificial sub-sampling.
We applied this statistical approach to two representative periods of the hydrological
cycle of the Orgeval Critical Zone Observatory: a typical 6-day rain event caused by the
arrival of a wet, Atlantic meteorological front (in October 2015) and a dry summer low
water stage period (July 2015) where the stream is essentially sustained by groundwater,
during an apparently steady hydrological period. We first present the behaviour of
calcium and sulphate concentrations as an example during the two considered periods
(Fig. 5 and 6), before generalizing to all measured species (Supplementary information
and Fig. SI 3, SI 5 and SI 6).

**Rain event**. The Ca concentration time series recorded at a 40-minutes frequency shows
that minimum Ca concentrations are recorded at maximum water discharge, but this
relationship is invisible at lower sampling frequency (Fig. 5). Narrow peaks during the
maximum of the stream flow are unresolved at a daily or 7-hourly frequency. The
comparison of the calculated PDF shows that a bimodal character is captured at all
frequencies. The average and standard deviation are not significantly affected by the
sampling frequency, with a relative difference of less than 2% for the values of these
parameters between the three distributions. However, the skewness values vary among
the different records. From the 40-minutes frequency to the daily frequency signals, the
skewness is weaker, which means that even if the overall concentration variability is
well captured at the lower sampling frequencies, the concentration signal is clearly
degraded. This degradation is particularly intense during the middle of the rain event,
where the concentration signal evolves quickly.

**Summer event.** Despite the absence of rain events during the 2015 summer, the River
Lab recorded high frequency variations revealing a diurnal structure with 7% relative
variations between day and night. Each element exhibits its own type of daily variation
in terms of amplitude and regularity. The Figure 6 shows that the structure of this signal
is altered when the sampling frequency decreases. While these daily variations are still
captured when sampling occurs every 7 hours, their amplitude is somewhat altered (5%)
compared to the 40-minutes sampling frequency (8%). The daily variability of the
signal is absent on the daily sampling frequency. While the mean remains the same over
the range of sampling frequency, the variability quantified by the relative standard
deviation decreases with lower sampling frequency, by up to 50% for the daily
frequency compared to the 40-minutes frequency signal, indicating a significant loss of
information. The skewness of the concentration distribution recorded at a sub-sampled
daily frequency has a value that is opposite in sign compared to the other two
frequencies, indicating that there is an inversion of the measured asymmetry of the PDF
at lower sampling frequencies. Therefore, too coarse of a sampling frequency can yield
a strongly altered signal compared to higher frequencies, resulting in a biased shape of
the distribution of the concentrations.

**Generalization.** The resampling approach applied above is generalized and expanded to
other elements for both the summer and rain events. The generalization to all species
measured is presented in supplementary information. In Figures 5 and 6, we arbitrarily
chose the hour of sampling (10 a.m. and 2 p.m. for Figures 5 and 6, respectively). In
figure SI 3, SI 5 and SI 6, the sub-sampling is performed at each of the possible
sampling hours. This statistical analysis quantitatively demonstrates that such high
frequency measurements are able to capture the day-night chemical cycles of the
Orgeval River. Given the amplitude and duration of typical rain events in the catchment,
the alteration of the signal by lowering the sampling frequency is less critical but still
significant during these periods (Supplementary information; Fig. SI 3, SI 5 and SI 6).

**5.3 What is revealed by better analytical precision?**
As shown above, the Orgeval RL not only achieves high-frequency measurements but
also results in improved precision compared to conventional lab analysis following
manual sampling. Therefore, any sampling procedure, even at a high frequency,
involving conventional lab analysis induces a loss of precision. We demonstrate this
effect through a numerically generated artificial degradation of the precision. Using the
original RL concentration signal as a reference, we artificially degraded the signals by
adding a normally distributed noise onto the concentration signals recorded by the RL.
Noise levels of 4% and 2% were tested as they are representative of the "standard"
analytical precision reported for most laboratory IC devices. The same representative
periods as in the previous section (summer and rain events) were utilized for these tests.
In this section we present the example of one element for each characteristic period
($Ca^{2+}$ for rain event Fig. 7 and $SO_4^{2+}$ for summer event Fig. 8. The generalization for all
elements is detailed in the supplementary information section (Supplementary
information and Fig. SI 4, SI 7 and SI 8).

**Rain event.** The Figure 7 illustrates the concentration PDF obtained after degradation
of the analytical precision for the Ca concentration. The narrow peaks recorded during
the maximum of the stream flow are virtually invisible in the signal at a 4%-precision,
and strongly smoothed in the signal at a 2%-precision. The original bimodal
characteristic of the PDF is still visible in the 2%-precision signal but no longer in the
4%-precision signal. The mean and standard deviation appear to be insensitive to these
changes in analytical precision, while the skewness is strongly impacted, reflecting
significant alteration of the concentration PDF at lower precision.

**Summer event.** Figure 8 shows how the sulphate concentration signal is affected when
the precision is degraded. Day-night variations are only visible in the original RL signal
because of its high analytical precision. The effect of degraded precision on the PDFs is
more important than for the rain event (Fig. 7). While the mean value is robust, the
standard deviation is altered (+150% from the RL signal to the 4% precision signal).
The skewness decreases (but keeps the same sign) by up to 90% for the signal at 4%-
precision compared to the original signal and 74% for the signal at 2%-precision,
indicating that the original RL signal asymmetry is lost as precision is worsened. These
changes in the parameters of the concentration PDF show that the structure of the
concentration signal in the Orgeval River would be significantly altered if the
measurements were made with analytical precision lower than that of the RL prototype.

**Generalization**. This approach has been expanded to other elements for both the
summer and rain events, as shown in the supplementary information, confirming that
concentration PDFs are strongly sensitive to the analytical precision for all species (Fig.
SI 4, SI 7 and SI 8).

**6 Conclusion**
This paper demonstrates the feasibility of deploying conventional laboratory
instruments in the field to measure the concentration of major dissolved anions and
cations in rivers ($Na^+$, $K^+$, $Mg^{2+}$, $Ca^{2+}$, $Cl^-$, $SO_4^{2-}$, $NO_3^-$) at a high frequency (one
measurement every 40 minutes) and at a high analytical precision (better than 1%) over
several months. The River Lab prototype was installed in the Avenelles stream at the
Orgeval Critical Zone Observatory, France. The RL features physico-chemical probes,
an on-line 0.2-μm pore size filtration system, and two ionic chromatographic devices,
all installed in a closed, air-conditioned bungalow. The RL is autonomous, remotely
operable and data can be transmitted automatically. Human intervention is required only
once a week. Therefore, the RL also allows for an efficient attribution of human
resources, as well as considerable saving of consumables.
A suite of tests performed on the RL to assess quality measurement and to compare
with more conventional "grab sampling" followed by laboratory measurements revealed
only a minor drift in the instrument calibration, leading to improved precision. This
precision is not easily achieved in the laboratory under standard analysis conditions,
showing the benefit of transporting the laboratory devices to the field. The analytical
capabilities of the RL for major dissolved elements could theoretically be extended to
other elements separable by ion chromatography. Preliminary tests demonstrate that
species present in trace amounts in river water (down to the ppb, such as strontium or
lithium) could be measured with the same gain in precision.
For this particular prototype, the measurement frequency (every 40 minutes) appears to
be limited by the turnover time of water in the filtered water circuit, which is itself
imposed by the filtration unit. However, the high frequency and high precision of the
RL enabled precise and accurate observations on the fine structure in hydrochemical
time series. Their interpretation is beyond the scope of the present proof-of-concept
paper but the RL is able to capture the abrupt changes in dissolved species
concentrations during a typical 6-days rain event, as well as daily oscillations during a
hydrological steady period of summer drought.
Using the high frequency RL signal as a benchmark, it is possible to artificially alter the
sample frequency and the analytical precision and study the resulting effect on the
hydrochemical distribution obtained for characteristic hydrological events. This analysis
shows that in order to retrieve the fine structure of the hydrochemical signal, high
sampling frequency and improved analytical precision are both necessary conditions. To
paraphrase James Kirchner's quote: "If we want to understand the full symphony of
catchment hydrochemical behaviour, then we need to be able to hear every note"
(Kirchner et al., 2004). The improvements made possible by the RL here or
concomitantly by von Freyberg et al. (2017) allow us to consider hearing the full
potamological symphony.
Future work will explore the relationships between the desired measurement frequency
and the timescales characterizing the complex interactions between primary and
secondary minerals, biotic processes and hydrological processes in catchments.
Recording such fine stream hydrochemical variations has the potential to offer a new
perspective in Critical Zone Science development.

**Author's information**
Corresponding author: *E-mail: floury@ipgp.fr and gaillardet@ipgp.fr

**Acknowledgment**
This work was supported by the EQUIPEX CRITEX programme, (grant # ANR-11-
EQPX-0011, PIs J. Gaillardet and L. Longuevergne) and funding from IRSTEA
(Institut Institut national de Recherche en Sciences et Technologies pour
l'Environnement et l'Agriculture). We thank Magadalena Niska for administrative help.
We would like to thank X. Zhang, Q. Charbonnier, D. Calmels, P. Louvat, J. Kirchner,
J. Druhan, S. Brantley, B. McDowell and J. Chorover for their help in the field and
helpful comments. A. Guerin (IRSTEA), S. Losa (Thermo Fisher), C. Fagot, P. Reignier
and M. Bauer from Endress+Hauser Company are thanked for technical assistance. PF
benefited from a doctorate grant from MESR, France. The Orgeval CZO river basin
belongs to the French National Infrastructure OZCAR (Observatoires de la Zone
Critique, Applications et Recherche).

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

**Table Captions**

**Table 1. Assessment of the RL accuracy and instrumental drift based on concentration measurements made after several injections of the standard solution "River x1". The uncertainty on the calibration solution is the quadratic sum of the uncertainty on the standard solutions (provided by the manufacturer) and the overall uncertainty for weighing during solution preparation. Measurement errors over one week and over two months are expressed as the relative standard deviation (RSD) calculated for repeated injections of the solution "River x1" directly into the IC instruments via the multiport valve (see Fig. 1).**

**Table 2. Precision on concentration measurements of the whole RL system calculated as the relative standard deviation (RSD) of concentration measurements made over three 24-hour closed loop experiments, during which the inlet and the outlet of the primary circuit are connected through a 300-L tank of river water.**

**Figure Captions**

**Figure 1. Sketch of the Orgeval River Lab. Bold blue arrows indicate the primary circuit of unfiltered water. Dashed arrows indicate filtered water supplied to IC instruments. 1: The inlet of the primary circuit samples the river at a constant 20-cm depth maintained by buoys. Water is first filtered through a < 2 mm pore size strainer. The distance between the mouth and the pump is 6 m. The primary circuit assembly is almost entirely composed of polyvinyl chloride (PVC) pipes. 2: The electric pump runs continuously at a constant power, leading to a rate of 700 liters per hour. 3: Almost all the river water just flows through the pipe and remains unfiltered. A fraction is filtered through a 2 μm tangential stainless steel filtration unit, then filtered through a 0.2 μm cellulose acetate frontal filter prior to being delivered to IC instruments at a flow rate of 1 liter per hour. 4: A multiport valve before introduction to the IC instruments allows for switching between filtered river water and standard or blank solutions. 5: All probes are deployed in an overflow tank of 5 liters of unfiltered river water. 6: The outlet of the primary circuit is downstream in the river.**

**Figure 2. Assessment of the precision (in deviation from the mean for 4 dissolved species) of the whole RL system including the primary circuit, filtration systems and IC instruments (April, 17th, 2016). A closed system is established on the primary circuit of the RL by connecting the inlet and the outlet through a 300-L tank of river water. The system is then run for a period of 24 hours. The time between two IC analyses is 40 minutes. The purple curve represents data of temperature of the water in the tank. We do not consider the 2 first hours (3 first measurements), corresponding to the homogenization of water in the circuit and tank (see conductivity measurements in Fig. SI 2) for the calculation of precision.**

**Figure 3. Cross-contamination assessment and response time of the RL system after a spike injection of 200 of NaCl. A closed system is established on the primary circuit of the RL by connecting the inlet and outlet through a 300-L tank of river water prior to the injection. The conductivity measurement frequency is 1 per minute, whereas the time between two measurements of chloride concentration is 40 minutes. Error bars for conductivity and Cl⁻ concentration measurements are within symbols size. Results are normalized to the difference between the minimum value, before the tracer injection (0%) and the maximum value, at the end of the experiment (100%).**

**Figure 4. Reproducibility assessment of IC measurements made by the RL every 40 minutes (blue), compared with concentration measurements made in the laboratory after conventional hourly river sampling (orange). Tests were performed on July 21st, 2015 and April 19th, 2016 for the cationic and ionic species respectively. For measurements performed in the laboratory, the error measurement is 1% (except for $K^+$ at 2%) calculated as the standard deviation**

over repeated injection of the standard solutions "River x1". For RL
measurements the error is given in Table 2.
**Figure 5. Calcium concentration and stream flow in the Orgeval river during a**
**rain event (from 1 to 25 October 2015), sampled every 40 minutes (RL original**
**signal at 40-minutes frequency) and artificially sub-sampled every 7 hours and**
**every day at 10 a.m. Black dots represent data during the rain event strictly (from**
**5 to 10 October 2015 at 10 a.m.), over which probability density functions (PDFs)**
**of concentration are calculated and represented as histograms (right panels). For**
**each PDF, the following statistical parameters are calculated: average (Ave.),**
**standard deviation (Std D.), and skewness (Skew.). Gray dots represent**
**concentration values outside of the rain event and are not considered in the**
**corresponding PDF. The two statistical parameters standard deviation (Std D.)**
**and skewness (Skew.) are not calculated for the daily subsampling because of the**
**too small number of points.**
**Figure 6. Sulphate concentration in the Orgeval river during a summer event**
**(from 7 to the 19 July 2015) sampled every 40 minutes (RL original signal) and**
**artificially sub-sampled every 7 hours, and every day at 2 p.m.. Probability density**
**functions (PDF) of concentration are represented as histograms (right panels). For**
**each PDF, the following statistical parameters are calculated: average (Ave.),**
**standard deviation (Std D.), and skewness (Skew.).**
**Figure 7. Calcium concentration and stream flow in the Orgeval river during a**
**rain event (from 1 to the 25 October 2015), as recorded by RL and for two**
**artificially degraded signals using a normally distributed noise with standard**
**deviation of 2% and 4%, to reflect the effect of decreased analytical precision.**
**Black dots represent data during the rain event strictly from 5 (12 a.m.) to 10**
**October 2015. The probability density functions (PDF) of concentration are**
**calculated and represented as histograms (right panels). For each PDF, the**
**following statistical parameters are calculated: average (Ave.), standard deviation**
**(Std D.) and skewness (Skew.). Gray dots represent concentration values outside of**
**the rain event, which are not considered for the analysis presented on the right**
**panels.**
**Figure 8. Sulphate concentration in the Orgeval river recorded by the RL during**
**two weeks in summer (7 to 19 July 2015), and for two artificially degraded signals,**
**using a normally distributed noise with a standard deviation of 2% and 4%, to**
**reflect the effect of degraded analytical precision. The probability density**
**functions (PDF) of concentration are calculated and represented as histograms**
**(right panels). The average (Ave.), standard deviation (Std D.), and skewness**
**(Skew.) are calculated for each PDF.**

Figure 1

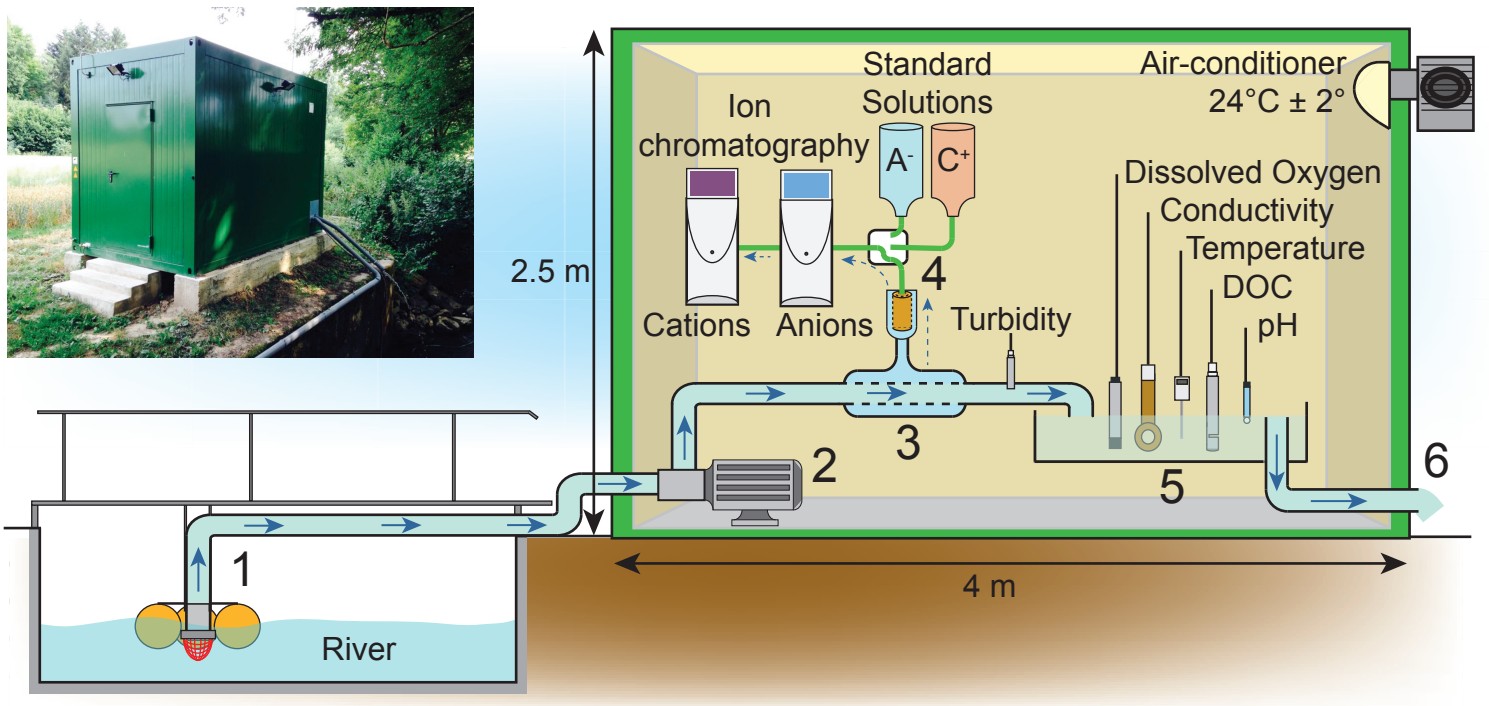

Ion chromatography
Standard Solutions
Air-conditioner 24°C ± 2°

A⁻  C⁺

Dissolved Oxygen
Conductivity
Temperature
DOC
pH

2.5 m

Cations   Anions   Turbidity

River

4 m

Figure 2

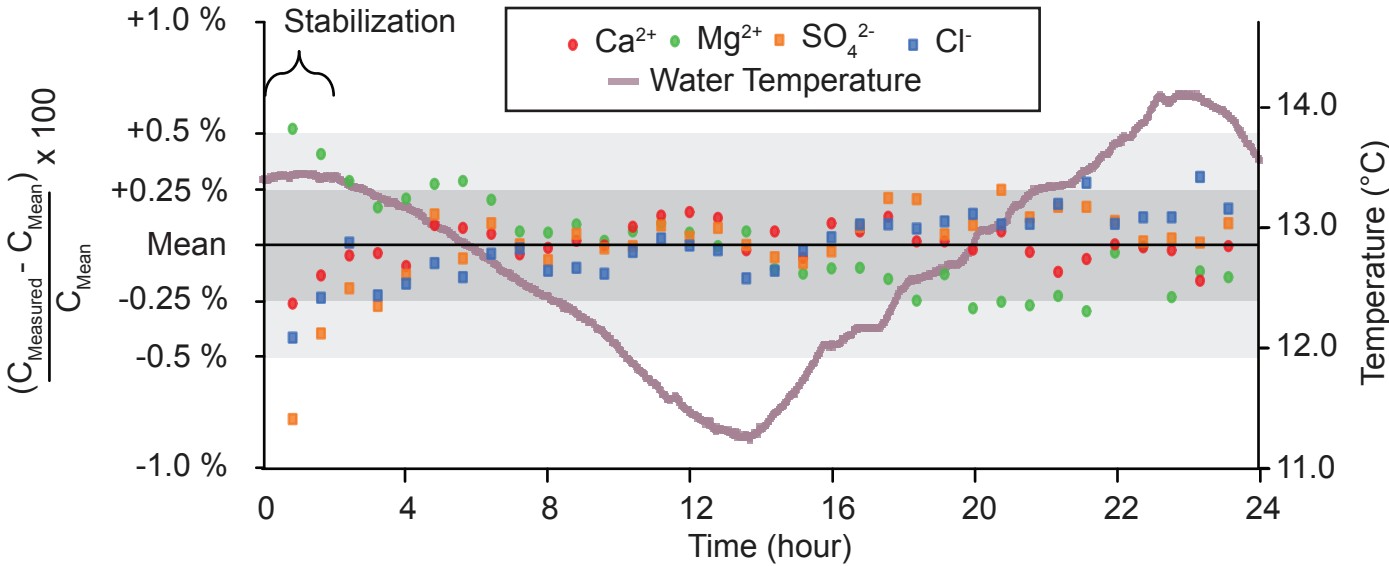

Figure 3

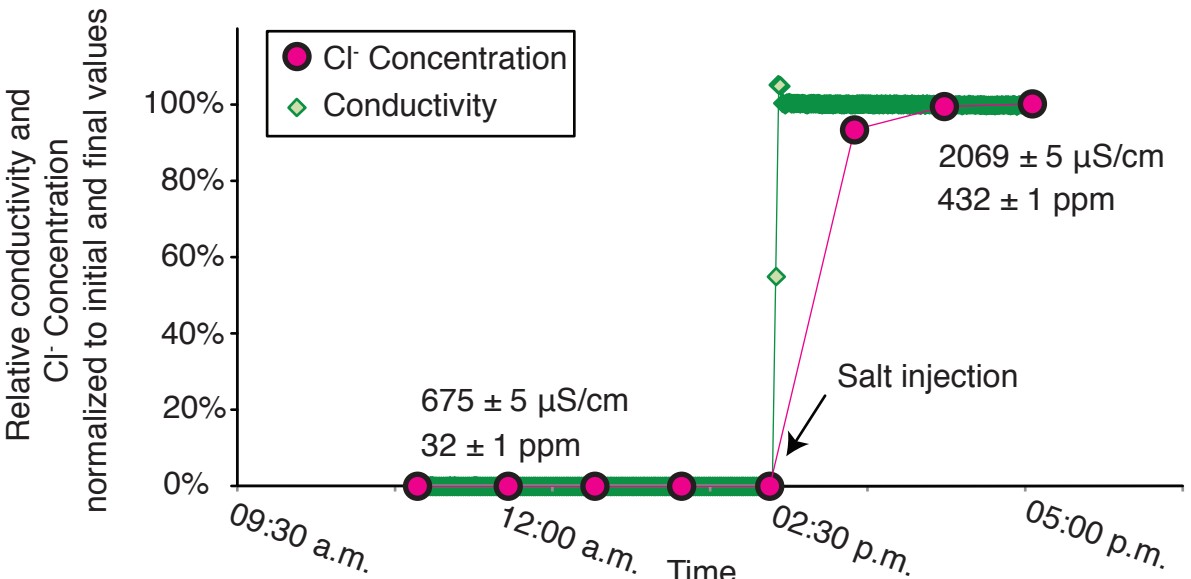

Cations (21-07-2015)
in ppm

Anions (19-04-2016)
in ppm

Ca²⁺

Na⁺

K⁺

Mg²⁺

NO₃⁻

SO₄²⁻

Cl⁻

Time

Time

● Orgeval River Lab
● IPGP Laboratory

Figure 5

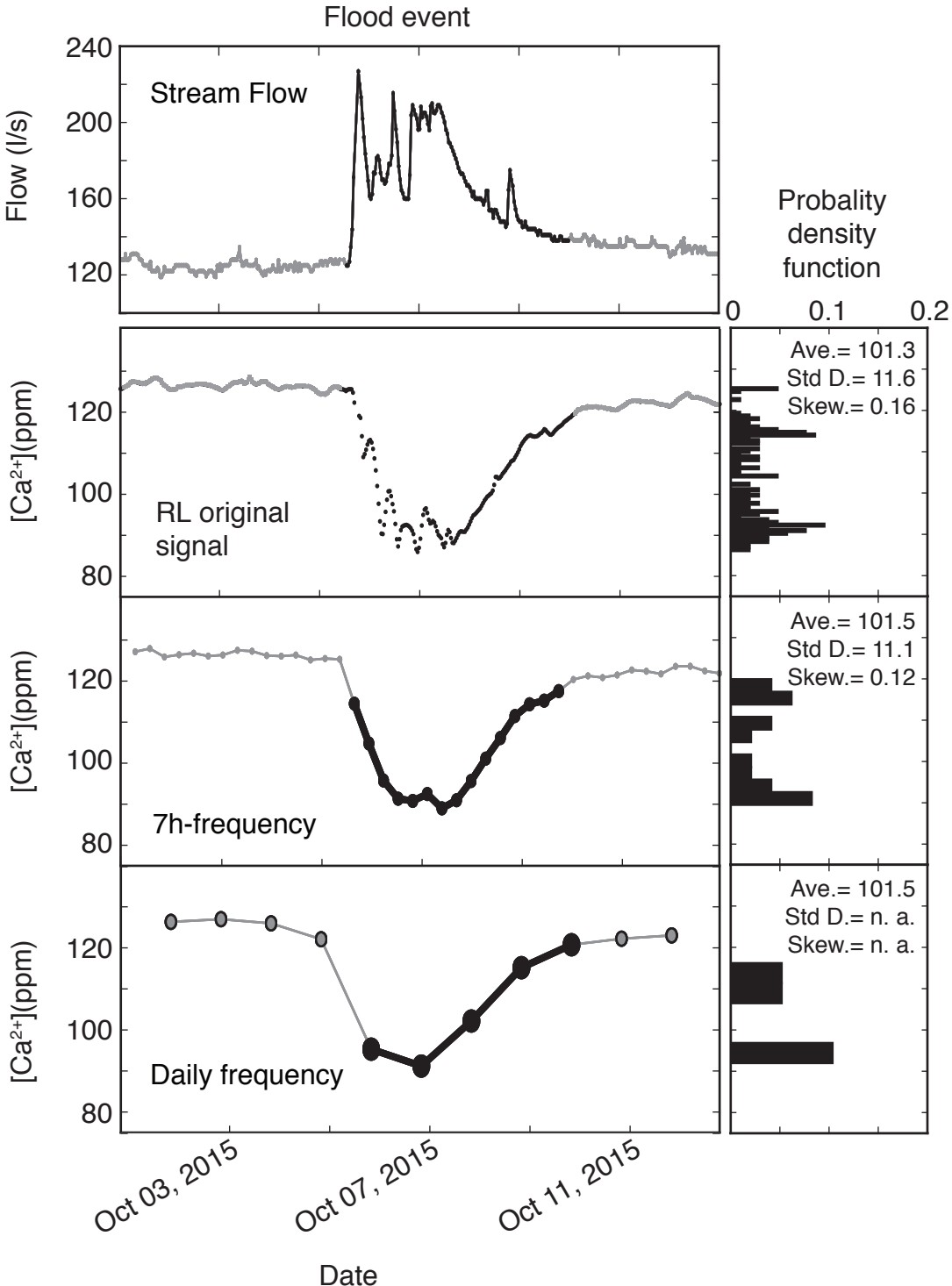

Summer event

Probality density function

**40-min frequency (RL original signal)**

Ave.= 68.7
Std D.= 1.37
Skew.= 1.04

**7-hourly frequency (sub-sampled)**

Ave.= 68.7
Std D.= 1.23
Skew.= 0.95

**Daily frequency (sub-sampled at 2p.m.)**

Ave.= 68.7
Std D.= 0.72
Skew.= -0.15

Figure 6

Date

Figure 7

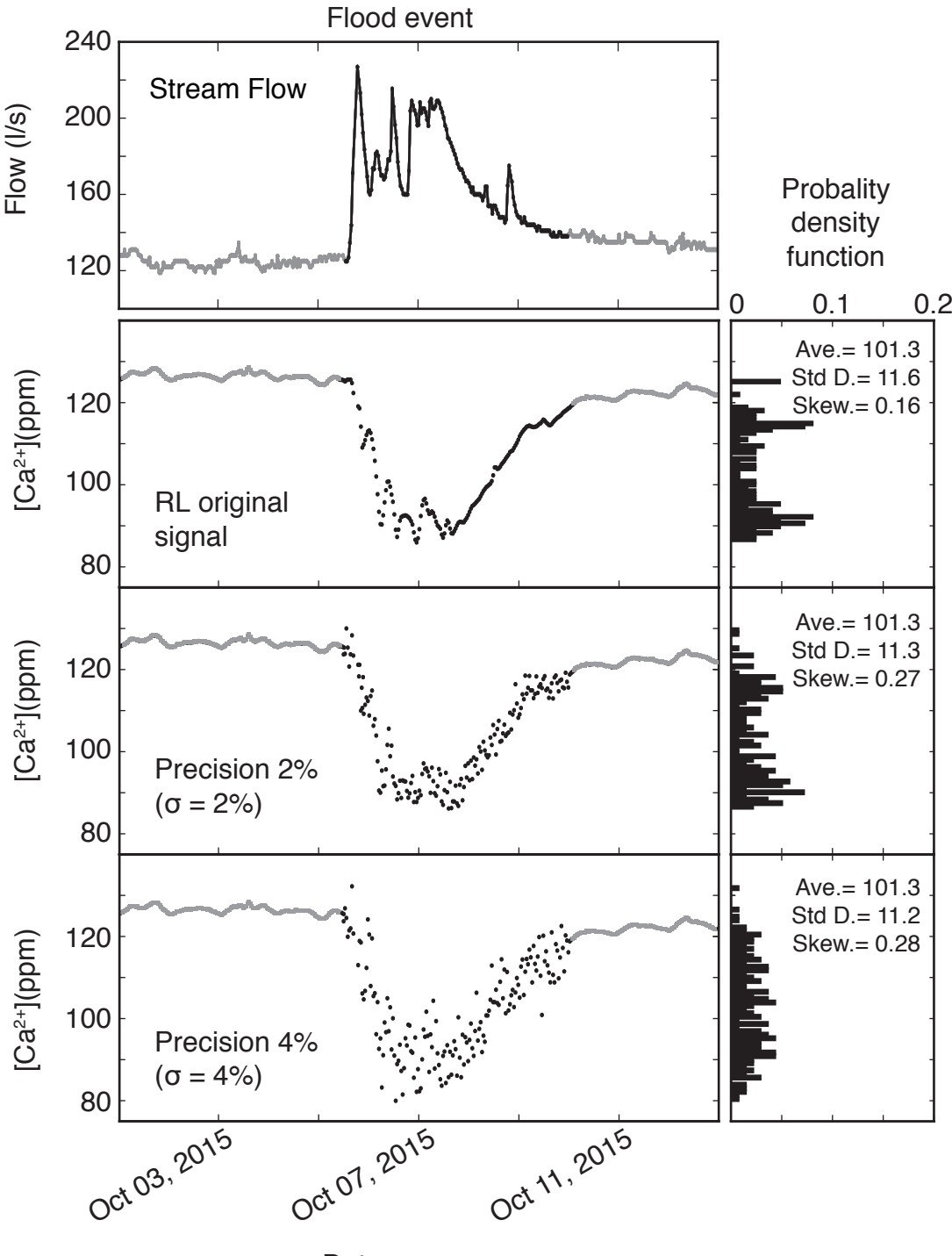

Date

Figure 8

Table 1

| | $Mg^{2+}$ | $K^+$ | $Ca^{2+}$ | $Na^+$ | $SO_4^{2-}$ | $NO_3^-$ | $Cl^-$ |
|---|---|---|---|---|---|---|---|
| Calibration Concentration | 10.0 | 3.0 | 130.0 | 10.0 | 70.0 | 60.0 | 40.0 |
| Uncertainty (mg.L$^{-1}$) | 0.03 | 0.01 | 0.39 | 0.03 | 0.84 | 0.84 | 0.28 |
| **Uncertainty (%)** | **0.3** | **0.45** | **0.3** | **0.3** | **1.2** | **1.4** | **0.7** |
| | | | | | | | |
| **One Measurement (Injection of "River x1" solution 4 times succsessivly)** | | | | | | | |
| Number of measurements | *(4)* | *(4)* | *(4)* | *(4)* | *(4)* | *(4)* | *(4)* |
| Average (mg.L$^{-1}$) | 10.08 | 3.00 | 129.86 | 9.98 | 70.26 | 60.31 | 40.32 |
| SD (mg.L$^{-1}$) | 0.02 | 0.01 | 0.16 | 0.02 | 0.69 | 0.63 | 0.27 |
| **RSD (%)** | **0.16** | **0.27** | **0.12** | **0.21** | **0.86** | **0.74** | **0.33** |
| | | | | | | | |
| **One Week (Injection of "River x1" solution every 8h)** | | | | | | | |
| Number of measurements | *(19)* | *(19)* | *(19)* | *(19)* | *(19)* | *(19)* | *(19)* |
| Average (mg.L$^{-1}$) | 10.13 | 3.02 | 130.64 | 10.01 | 70.54 | 60.63 | 40.44 |
| SD (mg.L$^{-1}$) | 0.03 | 0.01 | 0.39 | 0.02 | 0.67 | 0.44 | 0.22 |
| **RSD (%)** | **0.28** | **0.32** | **0.30** | **0.22** | **0.96** | **0.72** | **0.54** |
| | | | | | | | |
| **Two months (Injection of "River x1" solution every 2 days)** | | | | | | | |
| Number of measurements | *(28)* | *(28)* | *(28)* | *(28)* | *(25)* | *(25)* | *(25)* |
| Average (mg.L$^{-1}$) | 10.33 | 3.14 | 134.34 | 10.05 | 70.05 | 62.33 | 40.57 |
| SD (mg.L$^{-1}$) | 0.06 | 0.04 | 0.80 | 0.05 | 1.17 | 0.55 | 0.43 |
| **RSD (%)** | **0.54** | **1.34** | **0.59** | **0.50** | **1.68** | **0.92** | **1.07** |

Table 2

| Date | Number of measurements | $Mg^{2+}$ | $K^+$ | $Ca^{2+}$ | $Na^+$ | $SO_4^{2-}$ | $NO_3^-$ | $Cl^-$ |
|---|---|---|---|---|---|---|---|---|
| | | | | RSD (%) | | | | |
| 20th July 2015 | (22) | 0.17 | 0.90 | 0.21 | 0.22 | 0.39 | 0.47 | 0.24 |
| 28th August 2015 | (20) | 0.32 | 0.63 | 0.31 | 0.36 | 0.20 | 0.25 | 0.19 |
| 17th April 2016 | (35) | 0.38 | 1.20 | 0.17 | 0.31 | 0.31 | 0.38 | 0.30 |