# Peer review of "The Potamochemical symphony: new progress in the high-frequency acquisition of stream chemical data"

_Hydrology and Earth System Sciences, 2017_

## Referee Comment (RC1) · Anonymous Referee #1 · 23 Feb 2017

General comments

This proof-of-concept paper presents a new system of water chemistry monitoring (that the authors have termed 'River Lab') that offers both higher frequency monitoring and higher analytical precision than existing approaches. The authors describe this as a technical breakthrough that opens a new era of investigation into the hydrochemical signal of rivers. I agree that this is an exciting development and that we do need, generally in earth sciences, higher temporal and spatial monitoring for new catchment functioning insights (something that the Plynlimon dataset, that the authors refer to, showed superbly well). So I think this River Lab system is heading us in the right direction.

[Figure]

First, I think the title is rather good! I must admit to not knowing, when I first read this, that potamology is the study of rivers, but now that I do, I have a) learnt something, and b) been impressed with a cool title. The paper in general was really nice to read. It is well-written, clear and accessible throughout. The introduction was especially well-written and had a good progression through explaining to the reader what the norm is in stream chemistry monitoring, and why it is important to do better. It is also well-referenced. The figures are largely of good quality (although with a couple of minor exceptions noted below).

Overall, I think it is a good paper. I have only one 'major' comment: Regarding the offsets between the two sets of data (RL and Lab) shown in Figure 4, they seem rather large. You should further discuss the implications of this. They are systematic for most of the species. As you state on line 261 and throughout the paper, the RL data is more precise than the lab data (this is good, and you demonstrate this with data), but are the RL data more accurate than the lab data or vice versa? How could you determine this? You don't discuss accuracy at all. What is the use of a precise instrument if its accuracy is not so good! How does the existence of these offsets propagate into data analysis and process understanding? Surely we should care about the absolute magnitudes as well as the variations in amplitude. Or are the variations in amplitude most important?

I have quite a few, more minor comments, and these are listed below.

Minor comments

Line 24 Extra "a" in the sentence

Line 32 Is drought actually 'boring', hydrologically-speaking? I don't think so! Maybe rephrase to 'low flows' or something similar.

Abstract Combine into one paragraph

Line 48 Need to include the page number in Kirchner et al 2004, where that quote was taken from

Line 41-42 Your references need to be in some order... ie alphabetical or chronological... (this comment applies throughout the manuscript)

Line 53 "such as a single storm events" singular or plural?

Line 60-63 Would be great to read here example(s) of discoveries that Neal et al (2012)'s monitoring made... what didn't we know before that this study demonstrated was possible to know with high frequency chemistry monitoring?

Line 62-63 Why are "act of discovery" and "manual approaches" in quotation marks? Are they quotes from Neal et al (2012)? If so, say so. Otherwise, they don't seem to need quotation marks... (ie. not jargon)

Line 76 Does "Temporal" need to be capitalised?

Line 77 Is there a website / papers you could refer to on the CRITEX program?

Line 85 "Unsuspected" seems strange. Perhaps "unexpected"?

Line 98 I guess the 45 km2 refers to the Avenelles sub-catchment, rather than the Orgeval watershed, but this is ambiguous from the sentence structure.

Line 99 Should be "on average"

Line 105 Are these spring flash flood events from snowmelt? It would be good to see a figure of streamflow and precipitation inputs for a typical year or years.

Line 112 Give full names of IPGP and IRSTEA

Line 139 "the chosen analysis time is 30 minutes". Why is this the 'chosen' frequency? Is this the fastest the cationic and anionic measurements can be made? From my understanding of the paragraph starting at line 115, the probes in the primary circuit could operate on a 1-2 minute basis. So is 30 (or 40) minutes or the complete analysis (including anionic and cationic) an arbitrary decision or is this the highest frequency this set-up can manage? Would sampling at a higher frequency be desirable? Please

elaborate on this.

Line 143 Is there a word missing here?

Section 3 I would like to see more comparison with normal laboratory design and protocols in Section 3. Eg. is the calibration and cleaning frequency in keeping with lab practices?

Line 155-162 Very cool!

Line 190 Perhaps better to say "<" rather than "better than", or give a range of %

Line 191 Earlier you refer to Table 1 as "Table 1" and now you refer to it as "Tab. 1". Be consistent.

Line 201 "Fig S1 2". I can only see a Fig SI 1 in the supplementary document. Have you mis-referenced this?

Section 4.3 Could you apply autocorrelation analysis to your long time series (from the field) to check these cross-contamination errors under low vs high flow conditions?

Line 234 "In the case"... change to "in this case".

Line 249 Why did you only pick two dates at low summer flow? Why not for a high flow day? It would be good to see such a test at high flow periods too. Would we expect to see the same reproducibility for high flow?

Figure 4 Font size and type are not consistent between the different sub-plots. Correct this.

Line 297 Would the temperature of a laboratory not have similarly well-maintained temperature? In my (admittedly, somewhat limited direct) experience, they normally do.

Line 329 I think you can remove the parantheses here "(an apparently 'boring' hydrological period)"

Line 330-331 Did you actually sample all species, so that you have the equivalent data in Fig5+6 for all species (not just calcium and sulphate)? Do the fluctuations in those species show the same relationship with discharge? I would suggest including those data in the supplementary information. Using the average, Std D, skewness and kurtosis is great as a comparison tool, but how about some metric for how well the different sampling frequencies reveal the fluctuations in relationship to the streamflow hydrograph? How did you decide that average, Std D, skewness and kurtosis are the best comparison tools?

Figure 6 Include a hydrogaph on here, like you did with Figure 5. Does it have the same diurnal variation as the sulphate shows?

Line 348-349 revealing a diurnal structure [in sulphate]? Did the other species also exhibit the same diurnal structure? What do you mean by "specific to each element"?

Figure 7 This is a nice figure with LOTS of information contained within it. It's quite hard to imagine how these time series might look (the equivalent of Fig5+6). So again, this would support including the other species information in the supplementary information.

Line 397-398 "artificially degraded the signals by adding a normally distributed noise" Nice idea!

Figure 8 Could you make these into line plots (i.e. connect your dots) so that we can see the noise chronologically? And also include the vertical lines (like in Figure 5) from the peak discharge down through the plots.

Figure 9 Again, make these into line plots? Also add a hydrograph.

Line 474 Is this a paraphrase or a direct quote?

Line 475-477 Nice punchline. But (and I apologize for being really pedantic here), the quote is to 'hear' the notes, not to play them. The stream is playing the notes. Your RL is therefore. . . a really sophisticated hearing aid. . .???! (Maybe a better simile needed).

---

## Referee Comment (RC2) · Anonymous Referee #2 · 25 Feb 2017

**General comments**

This manuscript presents first data from a field deployment of an instrument package ("River Lab" or "RL") for high-frequency analysis of natural streamwaters. The instrumentation includes dual-channel ion chromatography and various physical and electrochemical probes. Data are presented to illustrate the quality of the chemical time series that can be obtained.

This manuscript makes a useful contribution to the growing literature on field-based chemical analyses of natural waters. In particular, the data presented here show that on-line analyses can be much more precise than those based on conventional sampling with later analysis in the lab.

However, the manuscript's characterization of the River Lab as a "breakthrough" (three times!) is not appropriate. Many other studies have also deployed wet chemistry instruments in the field (see Rode et al. ES&T, 50, 10297, 2016, and similar reviews). For example, a Swiss research group has recently published (also in HESS) a field deployment of another on-line ion chromatography system. That system arguably goes beyond the one presented here (it also includes isotopic analysis, and analyzes both rainfall and streamwater, rather than just streamwater alone), but the Swiss group does not use self-congratulatory terms like "breakthrough" to describe their work.

Likewise the claim that "... the high frequency and high precision of the RL enabled unprecedented observations of the fine structure in hydrochemical time series" is exaggerated. Similarly detailed results have been obtained before for other analytes using other field instrumentation, as well as by the Swiss group for major ions using IC. The present manuscript does a good job of demonstrating the precision of the RL, and does so in greater detail than I have seen before. But the observations themselves (at least the ones that are actually shown here) do not merit superlative terms like "unprecedented".

Heroic claims like "Our study opens a new era of investigation into the fine structure of the hydrochemical signal in rivers" (line 478) will not create a favorable impression in the community of investigators who have already been working, in some cases for decades, on these questions. Similarly, claims like "Recording such fine stream hydrochemical variations is thus offering a new perspective on Critical Zone" (line 483) are unproven and should be removed, since no inferences about any catchment processes are derived in the present manuscript.

Several time series are shown to illustrate the results that can be obtained with the RL, and to illustrate the results that would be expected at lower sampling frequencies. This is fine as far as it goes, but given that the manuscript claims that the system has been measuring seven ions continuously for over a year, it seems strange that only a few days of data, for only two ions, are shown. Where is the rest of the data set, for

the rest of the ions? It is understandable that the authors want to defer the analysis of the longer-term data set for a later paper, but they should at least demonstrate its existence by showing it to the audience.

To quantify the effects of subsampling and sampling error, the manuscript calculates how they affect the first four moments (mean, standard deviation, skewness, and kurtosis) of the distribution of measured concentrations for two brief sampling periods. Despite the time and space devoted to this analysis (two entire figures and parts of four others), it is not well posed and adds little to the paper, for the following reasons:

1) The moments of concentration distributions are rarely subjects of interest, particularly over such short periods of time.

2) The distributions (particularly in Fig. 5) are sensitive to the interval of time that is considered. In particular, much comment is made about the "bimodal" distribution during the "flood event", but this is largely an artifact of the specific time interval that is chosen.

3) Skewness and kurtosis are mostly useful in characterizing uni-modal distributions, and their application to bimodal distributions is not very helpful.

4) Statistical moments calculated from only five data points (as in Fig. 5) really should not be taken seriously!

5) The error bars in Figures 7 and 10 are unrealistic estimates of the uncertainties in the data points, because they do not account for the rather strong autocorrelation in the time series.

6) Skewness and kurtosis are not ratio-scale variables (they do not have a natural zero), so calculating percentage changes in them makes no mathematical sense, for the same reason that a temperature increase from 1 to 21 degrees Celsius is not "an increase of 2,000 percent!"

7) The changes in distributions and statistical moments shown in Figures 8-10 are

**C3**

unsurprising to anyone with even a modest background in statistics, given that we are mixing the distributions of the original data with an assumed error distribution that has a mean, skewness, and kurtosis of zero, plus a standard deviation that is substantial compared with that of the original data.

8) Statements like "the average is not sensitive to analytical precision" (line 429) are self-evident (of course it isn't, as long as the added noise has a mean of zero!).

The time series plots show the effects of subsampling and added noise very clearly. The statistical analysis does not help (and is often misleading), and should be omitted.

The manuscript is mostly well written, but in some places the grammar and word usage need improvement. To take just the two examples that arise in the title itself: "progresses" is not a noun in English, and "Potamochemical" is not a word in any language. While one can appreciate the authors' creativity here, the use of a made-up word like "Potamochemical" will seem pedantic to many readers. A few of us may still recall that "potamology" means "study of rivers", but the usage of this term has been declining steadily for about 50 years and there is no compelling reason to revive it. Readers should not need to reach for their dictionaries to look up obscure Greek roots of words, just to understand the title of a scientific paper.

**Specific comments**

The abstract claims that the RL was deployed "for over a year of continuous analyses" but the conclusion refers to measurements made "over several months". Which is it? Please provide plots showing "over a year of continuous analyses" of all ions, to substantiate the claim made in the abstract.

The abstract says that the daily oscillations observed during the summer drought were "unexpected". Why were these oscillations unexpected, given that they have been reported in many previous papers, including several that are cited here?

The characterization of sampling frequency in millihertz (i.e., 40-minute sampling is

described as 0.42 mHz) is unhelpful and will strike many readers as pedantry. If one wants to speak in terms of frequency, a more natural time base is 1 day (40-minute sampling is 36/day, 7-hour sampling is 3.42/day) or 1 week (40-minute sampling is 252/week, 7-hour sampling is 24/week).

The "flood event" represents an increase in flow of only about 50% or so, and even the highest flow in figure 5 (about 200 L/s) is 50 times smaller than the reported peak flow of over 10 m3/s (see line 105). In what sense is it appropriate to call this a "flood event"?

Figure 1 is artistic but less informative than it should be. Please provide a proper schematic of the instrumentation instead. Readers should be able to build a working version of the instrumentation (or at least understand the challenges involved in doing so) from the information provided.

Figure 3 focuses the reader's attention on the conductivity data (which is not the subject here), because it is darker than the CI data. Plot the conductivity data in a light color and the CI data in a dark color, and connect the CI points. And don't show the artificial 0-to-100 scale; show the real concentrations and conductivities.

Figure 4 and associated text: why are cations shown for one date and anions shown for another date? Show all the ions for one date, so they can be compared. Also, the fonts in Figure 4 are inconsistent, and the placement of the panels is erratic. Didn't anyone notice this? And the two colors look the same in grayscale; again, didn't anyone check? Also, please show the real hour of the real day, not an artificial scale (I went crazy trying to figure out what was going on here, before I discovered that the scale was fake and the cations and anions have nothing to do with one another because they are on different days (but why does chloride stop before the other anions?).

The laboratory analyses were reportedly done on "IC devices similar to those installed in the RL", but what does "similar to" mean? Specify exactly what instruments those were, what columns were used, how old they were, and so on. This is important,

because the relatively poor performance of the laboratory analyses is attributed to the difference in sampling, rather than the difference in instrumentation. If the laboratory instruments are an earlier generation of IC, or are using older columns, or have not been maintained as well, or have not been operated as carefully, or, or, or... it is easy to see how the laboratory results could look poorer for many different reasons.

The obvious offset between the laboratory results and those from the RL are particularly concerning. What efforts have been undertaken to verify that these do not indicate an artifact in either the RL or the laboratory?

In Figure 8, add error to the whole time series, not just the arbitrarily defined "event" period.

**Details (a partial list)**

50: Rozemeijer, not Rozemeiler 51: Chapman et al is 1997, not 1996 (and it would be appropriate to cite Neal et al., Hydrological Processes 2531, 2013, here). 292: discretely, not discreetly many figures: probability, not "probality" (didn't anybody notice this?) 476: "the best orchestra available": clever phrase, but an unproven assertion and will be perceived as inappropriate bragging 477: "to play the potamological symphony": no, instruments \*record\* the symphony; nature plays it.

---

## Author Comment (AC1) · 28 Mar 2017

**Dear Editor,**

**We would like to thank the two referees for their comments and useful advice on our manuscript. We appreciate the rather positive feedbacks provided by both reviews.**

**In the following, we will answer to all the reviewer's comments and suggest modifications that address reviewer's remarks and hence should improve the manuscript. We think that most of the reviewer's concerns are relevant and that addressing them will result in a significant improvement of the manuscript. Regarding some of the other concerns, we provide below the reasons why we do not agree.**

**Reviewer 1:** General comments

This proof-of-concept paper presents a new system of water chemistry monitoring (that the authors have termed 'River Lab') that offers both higher frequency monitoring and higher analytical precision than existing approaches. The authors describe this as a technical breakthrough that opens a new era of investigation into the hydrochemical signal of rivers. I agree that this is an exciting development and that we do need, generally in earth sciences, higher temporal and spatial monitoring for new catchment functioning insights (something that the Plynlimon dataset, that the authors refer to, showed superbly well). So I think this River Lab system is heading us in the right direction.

First, I think the title is rather good! I must admit to not knowing, when I first read this, that potamology is the study of rivers, but now that I do, I have a) learnt something, and b) been impressed with a cool title. The paper in general was really nice to read. It is well-written, clear and accessible throughout. The introduction was especially well- written and had a good progression through explaining to the reader what the norm is in stream chemistry monitoring, and why it is important to do better. It is also well- referenced. The figures are largely of good quality (although with a couple of minor exceptions noted below).

Overall, I think it is a good paper. I have only one 'major' comment: Regarding the offsets between the two sets of data (RL and Lab) shown in Figure 4, they seem rather large. You should further discuss the implications of this. They are systematic for most of the species. As you state on line 261 and throughout the paper, the RL data is more precise than the lab data (this is good, and you demonstrate this with data), but are the RL data more accurate than the lab data or vice versa? How could you determine this? You don't discuss accuracy at all. What is the use of a precise instrument if its accuracy is not so good! How does the existence of these offsets propagate into data analysis and process understanding? Surely we should care about the absolute magnitudes as well as the variations in amplitude. Or are the variations in amplitude most important?

**Authors: Regarding the comparison between the River Lab (RL) and in-lab measurements, we emphasize that the accuracy of the RL is demonstrated (Section 4.1 of our manuscript) through the measurements of the synthetic river water samples "River x1". The corollary (given that the difference between the two sets of measurements is systematic and beyond analytical precision) is that the in-lab measurements are - marginally, the 1-to-3% observed difference being fairly small compared to precision reported elsewhere for similar measurements - inaccurate.**

**Such accuracy with the RL was most likely achieved because (1) the calibration curve of the RL was made from a series of solutions (dilutions of the "River x1" solution) having the same element ratios as the solution used for the accuracy test (the "River x1"**

solution); (2) the RL is continuously processing solutions with a similar matrix, thereby minimizing memory effects and cross-contamination that can compromise measurements if widely differing samples are run successively on the same instrument. These two conditions were not met with our in-lab IC instruments, where we used a series of calibration solutions having the same concentration for all elements, and for which the measurement sessions took place between other sessions with very different samples. This is likely to lead to inaccuracies. Of course, we think that the accuracy of in-lab measurements could be improved by matching the matrix of the standards and samples, and/or by temporarily dedicating an instrument to one given type of sample (e.g. all samples from the same river). However, such conditions are not reflective of the "classical use" of an IC instrument in a lab, and we thus believe that our comparison is fair.

In the revised version of the manuscript, we will state more clearly that the RL provides the set of accurate measurements, and that our comparison is rather a check that the observed *relative* variations over a day are repeatable with another instrument.

Reviewer 1:I have quite a few, more minor comments, and these are listed below.

Minor comments

Line 24 Extra "a" in the sentence

Authors: This will be corrected in the final manuscript

Reviewer 1: Line 32 Is drought actually 'boring', hydrologically-speaking? I don't think so! Maybe rephrase to 'low flows' or something similar.

Authors: We used this term (importantly, with quotation marks) to reflect the fact that the variations in concentration were unexpected, but we recognize now that this is a bit blunt. This will be changed.

Reviewer 1: Abstract Combine into one paragraph

Authors: This will be corrected in the manuscript

Reviewer 1: Line 48 Need to include the page number in Kirchner et al 2004, where that quote was taken from

Authors: We will include these page numbers in the manuscript.

Reviewer 1: Line 41-42 Your references need to be in some order... ie alphabetical or chronological. . . (this comment applies throughout the manuscript)

Authors: We will fix this issue when revising the manuscript.

Reviewer 1: Line 53 "such as a single storm events" singular or plural?

Authors: We will change this sentence to singular: "such as a single storm event".

Reviewer 1: Line 60-63 Would be great to read here example(s) of discoveries that Neal et al (2012)'s monitoring made. . . what didn't we know before that this study demonstrated was possible to know with high frequency chemistry monitoring?

**Authors: We propose to add the following sentence:" The high sampling frequency on a long time scale provides new insights into hydrogeochemical functioning and a novel resource for catchment modelling. "**

**Reviewer 1:** Line 62-63 Why are "act of discovery" and "manual approaches" in quotation marks? Are they quotes from Neal et al (2012)? If so, say so. Otherwise, they don't seem to need quotation marks. . . (ie. not jargon)

**Authors: These are quotes from Neal et al. (2012). We will add this in the text.**

**Reviewer 1:** Line 76 Does "Temporal" need to be capitalised?

**Authors: No, we will correct this in the text.**

**Reviewer 1:** Line 77 Is there a website / papers you could refer to on the CRITEX program?

**Authors: We will add the link in the manuscript: https://www.critex.fr.**

**Reviewer 1:** Line 85 "Unsuspected" seems strange. Perhaps "unexpected"?

**Authors: This will be change to "unexpected".**

**Reviewer 1:** Line 98 I guess the 45 km2 refers to the Avenelles sub-catchment, rather than the Orgeval watershed, but this is ambiguous from the sentence structure.

**Authors: Yes, it refers to the Avenelles sub-catchment. We will change the sentence to be clearer.**

**Reviewer 1:** Line 99 Should be "on average"

**Authors: This will be corrected.**

**Reviewer 1:** Line 105 Are these spring flash flood events from snowmelt? It would be good to see a figure of streamflow and precipitation inputs for a typical year or years.

**Authors: No, these are flash rain events. As we want to keep the article short and focused on methodological aspects rather than on the field site, we will simply provide a reference linked to an article with all information about the hydrology of the river.**

**Reviewer 1:** Line 112 Give full names of IPGP and IRSTEA

**Authors: IPGP is "Institut de Physique du Globe de Paris" and IRSTEA is "Institut national de Recherche en Science et Technologie pour l'Environnement et l'Agriculture". We will add these explanations during revision.**

**Review 1:** Line 139 "the chosen analysis time is 30 minutes". Why is this the 'chosen' frequency? Is this the fastest the cationic and anionic measurements can be made? From my understanding of the paragraph starting at line 115, the probes in the primary circuit could operate on a 1-2 minute basis. So is 30 (or 40) minutes or the complete analysis (including anionic and cationic) an arbitrary decision or is this the highest frequency this set-up can manage? Would sampling at a higher frequency be desirable? Please elaborate on this.

**Authors: In our prototype, the sampling frequency is limited by technical issues. This will be explained by adding the following sentence: "However, the performed tests show**

that the frequency for a complete analyse of cation and anion is actually limited by the filtration device" (particularly by the turnover of the water inside). (Line 241- 243). Therefore, this highest frequency is not an arbitrary decision but the highest frequency reachable with our prototype.

**Reviewer 1:** Line 143 Is there a word missing here?

**Authors: Yes, "water". We will correct this in the text.**

**Reviewer 1:** Section 3 I would like to see more comparison with normal laboratory design and protocols in Section 3. Eg. is the calibration and cleaning frequency in keeping with lab practices?

**Authors: This information (working conditions, quality control, instruments models, columns…) is already in the supplementary information. We will move it to the main text.**

**Reviewer 1:** Line 155-162 Very cool!

Line 190 Perhaps better to say "<" rather than "better than", or give a range of %

**Authors: Yes, this wording is awkward, and will be corrected accordingly.**

**Reviewer 1:** Line 191 Earlier you refer to Table 1 as "Table 1" and now you refer to it as "Tab. 1". Be consistent.

**Authors: This will be corrected in the final manuscript.**

**Reviewer 1:** Line 201 "Fig S1 2". I can only see a Fig SI 1 in the supplementary document. Have you mis-referenced this?

**Authors: Yes, we will correct this in the text.**

**Reviewer 1:** Section 4.3 Could you apply autocorrelation analysis to your long time series (from the field) to check these cross-contamination errors under low vs high flow conditions?

**Authors: It is a relevant remark. It is one of our ways of research ways planned to perform such analyses for future papers.**

**Reviewer 1:** Line 234 "In the case". . . change to "in this case".

**Authors: We will correct this in the text.**

**Reviewer 1:** Line 249 Why did you only pick two dates at low summer flow? Why not for a high flow day? It would be good to see such a test at high flow periods too. Would we expect to see the same reproducibility for high flow?

**Authors: We thought that the summer drought would be an interesting period to perform these tests in an "extreme" case (lowest variations observed) and to let the reader appreciate the performance of the prototype. We also performed this test during a flood event but we do not present the results here because analytical conditions at that time were not perfectly controlled, such that we do not deem the results to be publishable. However, first-order results from the test on the flood event are consistent with summer tests.**

**Reviewer 1:** Figure 4 Font size and type are not consistent between the different sub-plots. Correct this.

**Authors: We will correct this in the text (this comment echoes a comment of reviewer 2).**

**Reviewer 1:** Line 297 Would the temperature of a laboratory not have similarly well-maintained temperature? In my (admittedly, somewhat limited direct) experience, they normally do.

**Authors: The room temperature in our laboratory (and we believe, in most laboratories) is not as carefully checked as in the RL prototype (24±2ºC). However, both instruments are the same and have a thermostat.**

**Reviewer 1:** Line 329 I think you can remove the parantheses here "(an apparently 'boring' hydro- logical period)"

**Authors: We will correct this in the text. "Boring" will be rephrased too, according to another comment of the reviewer.**

**Reviewer 1:** Line 330-331 Did you actually sample all species, so that you have the equivalent data in Fig5+6 for all species (not just calcium and sulphate)? Do the fluctuations in those species show the same relationship with discharge? I would suggest including those data in the supplementary information. Using the average, Std D, skewness and kurtosis is great as a comparison tool, but how about some metric for how well the different sampling frequencies reveal the fluctuations in relationship to the streamflow hydrograph? How did you decide that average, Std D, skewness and kurtosis are the best comparison tools?

**Authors: Yes, in principle we could add the other elements' variation as supplementary information but actually this is the purpose of a future article entirely devoted to summer daily variations.**

**Our approach to describe quantitatively the impact of the sampling frequency and precision on the signal relies on characterizing the concentration PDFs. The first four statistical moments (average, Std D, skewness and kurtosis) are classically used to characterize PDFs in a number of fields. As for using other approaches than simply the PDF (relationship with discharge, or time-series analysis), this will be the scope of future papers, and we did not want to expand too much on these approaches here.**

**Reviewer 1:** Figure 6 Include a hydrogaph on here, like you did with Figure 5. Does it have the same diurnal variation as the sulphate shows?

**Authors: For technical reasons, for the moment the discharge signal we have at hand for the summer drought is in the limit of quantification, hence not publishable, and requires further refinement (importantly, the discharge relative variations are lower than a few %). We do not think adding this discharge record is essential to improve the paper.**

**Reviewer 1:** Line 348-349 revealing a diurnal structure [in sulphate]? Did the other species also exhibit the same diurnal structure? What do you mean by "specific to each element"?

**Authors: We propose to add the following sentence to be clearer: "Each element exhibits its own type of daily variation in terms of amplitude and regularity"**

**Reviewer 1:** Figure 7 This is a nice figure with LOTS of information contained within it. It's quite hard to imagine how these time series might look (the equivalent of Fig5+6). So again, this would support including the other species information in the supplementary information.

**Authors: We agree with this comment, which meets concerns raised by the reviewer 2 (see below). We think that the calculations performed and generalized on all elements measured are meaningful to the reader however we think that having the other detailed time series would simply be too much for such a paper. Our goal with figures 7 and 10 was exactly to summarize the information that would be contained in these numerous time-series plots, to allow the reader to understand this at a glance. In addition, presenting an exhaustive report of all "real" time series (original signals) will be the subject of future papers.**

**Reviewer 1:** Line 397-398 "artificially degraded the signals by adding a normally distributed noise" Nice idea!

Figure 8 Could you make these into line plots (i.e. connect your dots) so that we can see the noise chronologically? And also include the vertical lines (like in Figure 5) from the peak discharge down through the plots.

**Authors: We tried to add the vertical lines as eye guides but found that connecting all dots makes the figure unreadable.**

**Reviewer 1:** Figure 9 Again, make these into line plots? Also add a hydrograph.

**Authors: We found that connecting all dots makes the figure unreadable.**

**Reviewer 1:** Line 474 Is this a paraphrase or a direct quote?

**Authors: A direct quote. We will add the reference to Kirchner et al. 2004 with the corresponding line number.**

**Reviewer 1:** Line 475-477 Nice punchline. But (and I apologize for being really pedantic here), the quote is to 'hear' the notes, not to play them. The stream is playing the notes. Your RL is therefore. . . a really sophisticated hearing aid. . .???! (Maybe a better simile needed).

**Authors: This comment is also formulated by the reviewer 2, we propose to change the sentence word "play" by "hear" and the sentence: "The improvements made possible by the RL allows us to hear almost each note of the potamological symphony."**
* * *
**Reviewer 2:** General comments

**Reviewer 2:** This manuscript presents first data from a field deployment of an instrument package ("River Lab" or "RL") for high-frequency analysis of natural streamwaters. The instrumentation includes dual-channel ion chromatography and various physical and electrochemical probes. Data are presented to illustrate the quality of the chemical time series that can be obtained.

This manuscript makes a useful contribution to the growing literature on field-based chemical analyses of natural waters. In particular, the data presented here show that on-line analyses can be much more precise than those based on conventional sampling with later analysis in the lab.

However, the manuscript's characterization of the River Lab as a "breakthrough" (three times!) is not appropriate. Many other studies have also deployed wet chemistry instruments in the field (see Rode et al. ES&T, 50, 10297, 2016, and similar reviews). For example, a Swiss research group has recently published (also in HESS) a field deployment of another on-line ion chromatography system. That system arguably goes beyond the one presented here (it also includes isotopic analysis, and analyzes both rainfall and stream water, rather than just stream water alone), but the Swiss group does not use self-congratulatory terms like "breakthrough" to describe their work.

**Authors: We are of course aware of the studies conducted by the Swiss group. We found it very interesting. We have already mentioned in our manuscript. However one could state that our manuscript also "arguably goes beyond the" paper by the Swiss research group in terms of analytical performances and their interest for hydrochemistry**.

**Reviewer 2:** Likewise the claim that "... the high frequency and high precision of the RL enabled unprecedented observations of the fine structure in hydrochemical time series" is exaggerated. Similarly detailed results have been obtained before for other analytes using other field instrumentation, as well as by the Swiss group for major ions using IC. The present manuscript does a good job of demonstrating the precision of the RL, and does so in greater detail than I have seen before. But the observations themselves (at least the ones that are actually shown here) do not merit superlative terms like "unprecedented".

**Authors: We appreciate the general comments of reviewer 2. We agree that the vocabulary is somewhat exaggerated and we will tone it down. However we notice that the word "breakthrough" that reviewer 2 does not like is used by reviewer 1 in his/her review.**

**Reviewer 2:** Heroic claims like "Our study opens a new era of investigation into the fine structure of the hydrochemical signal in rivers" (line 478) will not create a favorable impression in the community of investigators who have already been working, in some cases for decades, on these questions. Similarly, claims like "Recording such fine stream hydrochemical variations is thus offering a new perspective on Critical Zone" (line 483) are unproven and should be removed, since no inferences about any catchment processes are derived in the present manuscript.

**Authors: We partly agree with the reviewer that the words need more attention but we really think that our study is providing new information on the fine structure of river chemistry over such a long period of time. It was not our intention to make the reader feels that nothing was done before our study. We will remove the most "heroic claims"**

**Reviewer 2:** Several time series are shown to illustrate the results that can be obtained with the RL, and to illustrate the results that would be expected at lower sampling frequencies. This is fine as far as it goes, but given that the manuscript claims that the system has been measuring seven ions continuously for over a year, it seems strange that only a few days of data, for only two ions, are shown. Where is the rest of the data set, for the rest of the ions? It is understandable that the authors want to defer the analysis of the longer-term data set for a later paper, but they should at least demonstrate its existence by showing it to the audience.

**Authors: We understand the reviewer's comment but we would like to emphasize that in this article we first want to present the "proof of concept". This paper is thus not aimed at presenting the whole data acquired in one year. As correctly guessed by the reviewer, we "want to defer the analysis of the longer-term data set for a later paper". We only**

selected carefully two characteristic periods to illustrate the importance of recording river chemistry at high frequency and with a good precision. Presenting results over longer periods may dilute our conclusions. As stated in the introduction (and well understood by the reviewer 1) our paper aims at presenting the feasibility of measuring river chemistry by transporting the lab instruments to the field, not "giving away" a full year of measurements.

Reviewer 2: To quantify the effects of subsampling and sampling error, the manuscript calculates how they affect the first four moments (mean, standard deviation, skewness, and kurtosis) of the distribution of measured concentrations for two brief sampling periods. Despite the time and space devoted to this analysis (two entire figures and parts of four others), it is not well posed and adds little to the paper, for the following reasons:

1) The moments of concentration distributions are rarely subjects of interest, particularly over such short periods of time.

Authors: This is a totally unsupported statement that would deserve to be developed to warrant a proper answer. But we emphasize that we use these moments as a comparison tool between signals (original and altered) covering the *same periods of time*. In addition, we are doing this not in the scope of retrieving information on actual processes (which is probably what the reviewer has in mind when he/she says "subjects of interest"), but just in a sort of sensitivity analysis.

Reviewer 2: 2) The distributions (particularly in Fig. 5) are sensitive to the interval of time that is considered. In particular, much comment is made about the "bimodal" distribution during the "flood event", but this is largely an artefact of the specific time interval that is chosen.

Authors: We totally agree. However we carefully chose the interval between the very start and the very end of the flood event in order to not "under sample" this food event. By choosing a larger interval we would have altered the distribution by adding values from the non-flood period. Actually, reading our manuscript again we do not find that "much comment" is made about this bimodal distribution. We also note that many of the flood events we observed since the RL has been started show several peaks, such that choosing another flood event would not result in a different interpretation.

Reviewer 2: 3) Skewness and kurtosis are mostly useful in characterizing uni-modal distributions, and their application to bimodal distributions is not very helpful.

Authors: It all depends on what the reviewer means by "useful". It is true that for multi-modal distributions, the interpretation of skewness and kurtosis is not straightforward, but it is so for average and standard deviation as well... Therefore, we recognize that the chosen statistical moments are not *perfect* metrics of the shape of such multi-modal distributions, but (1) they have the advantage of being estimable empirically, and (2) importantly they are used here only as a tool of *comparison* between different PDFs.

Reviewer 2: 4) Statistical moments calculated from only five data points (as in Fig. 5) really should not be taken seriously!

Authors: We agree with the reviewer, this is slightly misleading (in the sense that with such low sample numbers, the empirical estimators of the statistical moments have no

chance to be close to their actual values), and we therefore removed the estimated statistical moments from the strongly sub-sampled signals (1/day) in Figs 5 and 7.

**Reviewer 2:** 5) The error bars in Figures 7 and 10 are unrealistic estimates of the uncertainties in the data points, because they do not account for the rather strong autocorrelation in the time series.

**Authors: We agree, and we propose to remove these error bars, and to replace them by a "cloud of points" corresponding to each realization of the sub-sampled (Fig. 7) / degraded (Fig. 10) signals. This would be more reflective of what we have actually done.**

**Reviewer 2:** 6) Skewness and kurtosis are not ratio-scale variables (they do not have a natural zero), so calculating percentage changes in them makes no mathematical sense, for the same reason that a temperature increase from 1 to 21 degrees Celsius is not "an increase of 2,000 percent!"

**Authors: We understand the reviewer concern: unlike average (at least average of ratio scale variables, such as chemical concentrations) and standard deviation, skewness and kurtosis are not ratio scale variables. However, we note that the "0" of the reduced centred skewness scale is not arbitrary. But indeed the simple fact it can take negative values shows that it is not a ratio scale variable. Obviously, a possibility would have been to use the absolute value of skewness, but this would have been too difficult to follow. We therefore suggest that in the revised figures we will present the absolute values of these metrics in Figs. 7 and 10.**

**Reviewer 2:** 7) The changes in distributions and statistical moments shown in Figures 8-10 are unsurprising to anyone with even a modest background in statistics, given that we are mixing the distributions of the original data with an assumed error distribution that has a mean, skewness, and kurtosis of zero, plus a standard deviation that is substantial compared with that of the original data.

**Authors: We are not sure to understand the reviewer's point. Results presented in Figs. 8-10 show that the mean is not affected by adding this normal noise, but all three other moments are. We believe that this will not be easily predicted by readers having "a modest background in statistics". And let alone quantitatively predicted - which is another interest of this approach: actual numbers are given. For example, our initial guess would have been, if anything, that only the standard deviation would be affected, since the standard deviation of the added signal is its only non-zero statistical moment.**

**Reviewer 2:** 8) Statements like "the average is not sensitive to analytical precision" (line 429) are self-evident (of course it isn't, as long as the added noise has a mean of zero!).

**Authors: This is true and our intention was not to present this as a surprise. We will rephrase the sentence about the average.**

**Reviewer 2:** The time series plots show the effects of subsampling and added noise very clearly. The statistical analysis does not help (and is often misleading), and should be omitted.

**Authors: As explained above, we do not agree with the reviewer about the interest of the statistical analysis (which allows us to build Figs. 7 and 10, and therefore to generalize the approach to all elements and report the results in little space, which would be impossible using the time series only). Therefore we will leave this statistical analysis.**

**Reviewer 2:** The manuscript is mostly well written, but in some places the grammar and word us- age need improvement. To take just the two examples that arise in the title itself: "progresses" is not a noun in English, and "Potamochemical" is not a word in any language. While one can appreciate the authors' creativity here, the use of a made-up word like "Potamochemical" will seem pedantic to many readers. A few of us may still recall that "potamology" means "study of rivers", but the usage of this term has been declining steadily for about 50 years and there is no compelling reason to revive it. Readers should not need to reach for their dictionaries to look up obscure Greek roots of words, just to understand the title of a scientific paper.

**Authors: We can understand the reviewer's feeling, which is exactly the opposite of that of reviewer 1. It is a personal point of view. On our side, we do not think that, because a term is declining, it should fade away. There is no intent in trying to look smart in this article, but we think that the almost forgotten term "potamology" should be revived so it could increase the visibility of our river and critical zone communities. We are not considering changing this word. We will change "progresses" to "progress".**

**Reviewer 2:** Specific comments

The abstract claims that the RL was deployed "for over a year of continuous analyses" but the conclusion refers to measurements made "over several months". Which is it? Please provide plots showing "over a year of continuous analyses" of all ions, to substantiate the claim made in the abstract.

**Authors: We refer the reviewer to its own sentence above: we "want to defer the analysis of the longer-term data set for a later paper" in preparation. This is absolutely unnecessary for this "proof-of-concept" paper.**

**Reviewer 2:** The abstract says that the daily oscillations observed during the summer drought were "unexpected". Why were these oscillations unexpected, given that they have been reported in many previous papers, including several that are cited here?

**Authors: We will remove the word "unexpected".**

**Reviewer 2:** The characterization of sampling frequency in millihertz (i.e., 40-minute sampling is described as 0.42 mHz) is unhelpful and will strike many readers as pedantry. If one wants to speak in terms of frequency, a more natural time base is 1 day (40-minute sampling is 36/day, 7-hour sampling is 3.42/day) or 1 week (40-minute sampling is 252/week, 7-hour sampling is 24/week).

**Authors: We propose to change in the manuscript the frequencies with "40-minutes", "7-hourly" and "1/day". We propose to remove all terms in "hertz" as we realize that is note clear for the readers.**

**Reviewer 2:** The "flood event" represents an increase in flow of only about 50% or so, and even the highest flow in figure 5 (about 200 L/s) is 50 times smaller than the reported peak flow of over 10 m3/s (see line 105). In what sense is it appropriate to call this a "flood event"?

**Authors: Indeed. To be fairer, we will change the term "flood event" to "rain event".**

**Reviewer 2:** Figure 1 is artistic but less informative than it should be. Please provide a proper schematic of the instrumentation instead. Readers should be able to build a working version

of the instrumentation (or at least understand the challenges involved in doing so) from the information provided.

**Authors: We think that this figure is a very good compromise between readability, such that the average reader will understand at a glance what the equipment is about, while the more interested reader will obtain first-order technical information from it. We will therefore keep this figure in the main paper. However, we will add in the next version a more technical sketch with all information needed in the supplementary information.**

**Reviewer 2:** Figure 3 focuses the reader's attention on the conductivity data (which is not the subject here), because it is darker than the Cl data. Plot the conductivity data in a light color and the Cl data in a dark color, and connect the Cl points. And don't show the artificial 0-to-100 scale; show the real concentrations and conductivities.

**Authors: Absolute values for each measurement are already given in the figure. We still think that a normalized scale help the reader to focus on the relative delay to appreciate the cross-contamination and not the absolute value.**

**Reviewer 2:** Figure 4 and associated text: why are cations shown for one date and anions shown for another date? Show all the ions for one date, so they can be compared. Also, the fonts in Figure 4 are inconsistent, and the placement of the panels is erratic. Didn't anyone notice this? And the two colors look the same in grayscale; again, didn't anyone check? Also, please show the real hour of the real day, not an artificial scale (I went crazy trying to figure out what was going on here, before I discovered that the scale was fake and the cations and anions have nothing to do with one another because they are on different days (but why does chloride stop before the other anions?).

**Authors: We apologize for these mistakes, and all of the requested format changes will be carried out. We will also state more explicitly that these analyses were carried out over different days for anions and cations (which was necessary for logistical reasons, but which does not impair our approach).**

**Reviewer 2:** The laboratory analyses were reportedly done on "IC devices similar to those installed in the RL", but what does "similar to" mean? Specify exactly what instruments those were, what columns were used, how old they were, and so on. This is important, because the relatively poor performance of the laboratory analyses is attributed to the difference in sampling, rather than the difference in instrumentation. If the laboratory instruments are an earlier generation of IC, or are using older columns, or have not been maintained as well, or have not been operated as carefully, or, or, or... it is easy to see how the laboratory results could look poorer for many different reasons.

**Authors: As already answered to the reviewer 1, such information is already given in the supplement. We will move this information to the main text.**

**Reviewer 2:** The obvious offset between the laboratory results and those from the RL are particularly concerning. What efforts have been undertaken to verify that these do not indicate an artifact in either the RL or the laboratory?

**Authors: We invite the reviewer 2 to read our reply formulated earlier to the first paragraph of the review 1.**

**Reviewer 2:** In Figure 8, add error to the whole time series, not just the arbitrarily defined "event" period. Details (a partial list) 50: Rozemeijer, not Rozemeiler 51: Chapman et al is

1997, not 1996 (and it would be appropriate to cite Neal et al., Hydrological Processes 2531, 2013, here). 292: discretely, not discreetly many figures: probability, not "probality" (didn't anybody notice this?) 476: "the best orchestra available": clever phrase, but an unproven assertion and will be perceived as inappropriate bragging 477: "to play the potamological symphony": no, instruments *record* the symphony; nature plays it.

**Authors: We will add these new references and correct the typos. We also invite the reviewer 2 to see our reply to reviewer 1 regarding the mistake between "play" and hear".**

---

## Author Response (AR1)

**Dear Editor,**

**We present here a revised version of the manuscript "The Potamochemical symphony: new progress in the high-frequency acquisition of stream chemical data" according to the comments and advice from the two referees. We also correct several minor errors. All modifications are highlighted in the marked-up manuscript.**

**We would like to mention that the structure of the discussion has been reviewed. The two paragraphs called "generalization" for both part 5.2 and 5.3 are summarized in the manuscript and more information and the figures corresponding are now available in the supplementary information as suggested by both reviewers. (Please see Figure SI 3, 4, 5, 6, 7 and8).**

**We also change the figures as following:**

**Figure 7 has been move to supplementary information: Figure SI 3, 5 and 6**

**Figure 8 is now Figure 7**

**Figure 9 is now Figure 8**

**Figure 10 has been move to supplementary information: Figure SI 4, 7 and 8**

**We change several details and errors in the figures 1, 2, 3, 4, 5, 6, 7 and 8.**

**We also corrected some minor details in the Table 1 and 2.**

**In the supplementary information, we add a new figure presenting the technical plan of the RiverLab. Please see figure SI1.**

**In the following, we present the modifications for each remarks and comments that address reviewer's. We also provide a point-by-point reply to the comments with the modifications in the manuscript corresponding.**

**Reviewer 1:** General comments

This proof-of-concept paper presents a new system of water chemistry monitoring (that the authors have termed 'River Lab') that offers both higher frequency monitoring and higher analytical precision than existing approaches. The authors describe this as a technical breakthrough that opens a new era of investigation into the hydrochemical signal of rivers. I agree that this is an exciting development and that we do need, generally in earth sciences, higher temporal and spatial monitoring for new catchment functioning insights (something that the Plynlimon dataset, that the authors refer to, showed superbly well). So I think this River Lab system is heading us in the right direction.

First, I think the title is rather good! I must admit to not knowing, when I first read this, that potamology is the study of rivers, but now that I do, I have a) learnt something, and b) been impressed with a cool title. The paper in general was really nice to read. It is well-written, clear and accessible throughout. The introduction was especially well- written and had a good progression through explaining to the reader what the norm is in stream chemistry monitoring, and why it is important to do better. It is also well- referenced. The figures are largely of good quality (although with a couple of minor exceptions noted below).

Overall, I think it is a good paper. I have only one 'major' comment: Regarding the offsets between the two sets of data (RL and Lab) shown in Figure 4, they seem rather large. You should further discuss the implications of this. They are systematic for most of the species. As you state on line 261 and throughout the paper, the RL data is more precise than the lab data (this is good, and you demonstrate this with data), but are the RL data more accurate than the lab data or vice versa? How could you determine this? You don't discuss accuracy at all. What is the use of a precise instrument if its accuracy is not so good! How does the existence of these offsets propagate into data analysis and process understanding? Surely we should care about the absolute magnitudes as well as the variations in amplitude. Or are the variations in amplitude most important?

**Authors:**

**In the revised version of the manuscript, we have state more clearly that the RL provides the set of accurate measurements, and that our comparison is rather a check that the observed *relative* variations over a day are repeatable with another instrument. We add the following sentence:** *"In addition, the most accurate measurements were obtained with the RL ratther than with the laboratory equipment because (1) the calibration curve of the RL was made from a series of solutions (dilutions of the "River x1" solution) having the same element ratios as the solution used for the accuracy test (the "River x1" solution); (2) the RL is continuously processing solutions with a similar matrix, thereby minimizing memory effects and cross-contamination that can compromise measurements if widely differing samples are run successively on the same instrument. These two conditions were not met with our in-lab IC instruments, where we used a series of calibration solutions having the same concentration for all elements, and for which the measurement sessions took place between other sessions with very different samples. These features of the measurement protocol, representative of most laboratory workflows for hydrochemical measurements, are likely to lead to inaccuracies."* **Line 362 – Line 373.**

**Reviewer 1:**I have quite a few, more minor comments, and these are listed below.

Minor comments

Line 24 Extra "a" in the sentence

**Authors: This has been corrected in the final manuscript**

**Reviewer 1:** Line 32 Is drought actually 'boring', hydrologically-speaking? I don't think so! Maybe rephrase to 'low flows' or something similar.

**Authors: We change this term by** *"a hydrological low-flow period of summer drought."* **Line 26.**

**Reviewer 1:** Abstract Combine into one paragraph

**Authors: This has been corrected in the manuscript**

**Reviewer 1:** Line 48 Need to include the page number in Kirchner et al 2004, where that quote was taken from

**Authors: We include these page numbers in the manuscript. Line 50**

**Reviewer 1:** Line 41-42 Your references need to be in some order... ie alphabetical or chronological. . . (this comment applies throughout the manuscript)

**Authors: We fix this issue in the manuscript.**

**Reviewer 1:** Line 53 "such as a single storm events" singular or plural?

**Authors: We change this sentence to singular:** *"a single storm event"*. **Line 55**

**Reviewer 1:** Line 60-63 Would be great to read here example(s) of discoveries that Neal et al (2012)'s monitoring made. . . what didn't we know before that this study demonstrated was possible to know with high frequency chemistry monitoring?

**Authors: We add the following sentence: "** *The authors demonstrate the "act of discovery" permitted by such sampling scheme, as the high sampling frequency of river hydrochemistry over sufficiently long time spans revealed patterns related to hydrological and biological drivers invisible at lower sampling frequency."* **Line 80-83**

**Reviewer 1:** Line 62-63 Why are "act of discovery" and "manual approaches" in quotation marks? Are they quotes from Neal et al (2012)? If so, say so. Otherwise, they don't seem to need quotation marks. . . (ie. not jargon)

**Authors: These are quotes from Neal et al. (2012). We add this in the text. Line 80 and we remove the quote mark for manual approaches like from Neal et al (2012). Line 84.**

**Reviewer 1:** Line 76 Does "Temporal" need to be capitalised?

**Authors: No, we correct this in the text. Line 107**

**Reviewer 1:** Line 77 Is there a website / papers you could refer to on the CRITEX program?

**Authors: We add the link in the manuscript: https://www.critex.fr. Line 108**

**Reviewer 1:** Line 85 "Unsuspected" seems strange. Perhaps "unexpected"?

**Authors: This be remove the word to "***Unsuspected***". (Please see also reply to reviewer 2). Line 117**

**Reviewer 1:** Line 98 I guess the 45 km2 refers to the Avenelles sub-catchment, rather than the Orgeval watershed, but this is ambiguous from the sentence structure.

**Authors: Yes, it refers to the Avenelles sub-catchment. We change the sentence to be clearer: "***The Avenelles River drains an area of 45 km$^2$***.". Line 131**

**Reviewer 1:** Line 99 Should be "on average"

**Authors: This has been corrected. Line 149**

**Reviewer 1:** Line 105 Are these spring flash flood events from snowmelt? It would be good to see a figure of streamflow and precipitation inputs for a typical year or years.

**Authors: No, these are flash rain events. As we want to keep the article short and focused on methodological aspects rather than on the field site, we will simply provide a reference linked to an article with all information about the hydrology of the river. Catchment hydrologic data are available on the ORACLE website (https://bdoh.irstea.fr/ORACLE/). Line 128**

**Reviewer 1:** Line 112 Give full names of IPGP and IRSTEA

**Authors: We add the full name: "*The RL was designed by IPGP (Institut de Physique du Globe de Paris, France) and IRSTEA  (Institut national de Recherche en Sciences et Technologies pour l'Environnement et l'Agriculture, France)*". Line 161-164**

**Review 1:** Line 139 "the chosen analysis time is 30 minutes". Why is this the 'chosen' frequency? Is this the fastest the cationic and anionic measurements can be made? From my understanding of the paragraph starting at line 115, the probes in the primary circuit could operate on a 1-2 minute basis. So is 30 (or 40) minutes or the complete analysis (including anionic and cationic) an arbitrary decision or is this the highest frequency this set-up can manage? Would sampling at a higher frequency be desirable? Please elaborate on this.

**Authors: This has been explained by adding the following sentence: "*Our tests show that the frequency for a complete and uncontaminated analyse of cation and anion is actually limited by the filtration device (see part 4.3).*" Line 193-194**

**Reviewer 1:** Line 143 Is there a word missing here?

**Authors: Yes, "*water*". We correct this in the text. Line 150.**

**Reviewer 1:** Section 3 I would like to see more comparison with normal laboratory design and protocols in Section 3. Eg. is the calibration and cleaning frequency in keeping with lab practices?

**Authors: We add the following sentence "*Pure distilled water is regularly (every two weeks) introduced to check the residual noise.*" Line 202-203**

**Reviewer 1:** Line 155-162 Very cool!

Line 190 Perhaps better to say "<" rather than "better than", or give a range of %

**Authors: Yes, this wording is awkward, and has been corrected by "*lower*". Line 263**

**Reviewer 1:** Line 191 Earlier you refer to Table 1 as "Table 1" and now you refer to it as "Tab. 1". Be consistent.

**Authors: This has been corrected in the final manuscript with (Tab. 1). Line 251, 264 and 266**

**Reviewer 1:** Line 201 "Fig S1 2". I can only see a Fig SI 1 in the supplementary document. Have you mis-referenced this?

**Authors: We change the supplementary information, this is the Figure SI2 and corrected in the manuscript. Line 285**

**Reviewer 1:** Section 4.3 Could you apply autocorrelation analysis to your long time series (from the field) to check these cross-contamination errors under low vs high flow conditions?

**Authors: It is a relevant remark. It is one of our ways of research ways planned to perform such analyses for future papers.**

**Reviewer 1:** Line 234 "In the case". . . change to "in this case".

**Authors: We correct this in the text. Line 324**

**Reviewer 1:** Line 249 Why did you only pick two dates at low summer flow? Why not for a high flow day? It would be good to see such a test at high flow periods too. Would we expect to see the same reproducibility for high flow?

**Authors: We thought that the summer drought would be an interesting period to perform these tests in an "extreme" case (lowest variations observed) and to let the reader appreciate the performance of the prototype. We also performed this test during a flood event but we do not present the results here because analytical conditions at that time were not perfectly controlled, such that we do not deem the results to be publishable. However, first-order results from the test on the flood event are consistent with summer tests.**

**Reviewer 1:** Figure 4 Font size and type are not consistent between the different sub-plots. Correct this.

**Authors: We correct this in the figure. Please see Figure 4 (this comment echoes a comment of reviewer 2).**

**Reviewer 1:** Line 297 Would the temperature of a laboratory not have similarly well-maintained temperature? In my (admittedly, somewhat limited direct) experience, they normally do.

**Authors: The room temperature in our laboratory (and we believe, in most laboratories) is not as carefully checked as in the RL prototype (24±2ºC). However, both instruments are the same and have a thermostat.**

**Reviewer 1:** Line 329 I think you can remove the parantheses here "(an apparently 'boring' hydro- logical period)"

**Authors: We correct this in the text by "*an apparently steady hydrological period.*" Line 450. "Boring" was rephrased too, according to another comment of the reviewer; we remove this word in the manuscript.**

**Reviewer 1:** Line 330-331 Did you actually sample all species, so that you have the equivalent data in Fig5+6 for all species (not just calcium and sulphate)? Do the fluctuations in those species show the same relationship with discharge? I would suggest including those data in the supplementary information. Using the average, Std D, skewness and kurtosis is great as a comparison tool, but how about some metric for how well the different sampling frequencies reveal the fluctuations in relationship to the streamflow hydrograph? How did you decide that average, Std D, skewness and kurtosis are the best comparison tools?

**Authors: Our approach to describe quantitatively the impact of the sampling frequency and precision on the signal relies on characterizing the concentration PDFs. As recommended by the reviewer 2, we present in the supplementary information. We remove the Kurtosis and present the generalization to all elements in the supplementary information (Figure SI 3, 4, 5, 6, 7 and 8). Please see the reply to the reviewer 2 further.**

**Reviewer 1:** Figure 6 Include a hydrograph on here, like you did with Figure 5. Does it have the same diurnal variation as the sulphate shows?

**Authors: For technical reasons, for the moment the discharge signal we have at hand for the summer drought is in the limit of quantification, hence not publishable, and requires**

**further refinement (importantly, the discharge relative variations are lower than a few %). We do not think adding this discharge record is essential to improve the paper.**

**Reviewer 1:** Line 348-349 revealing a diurnal structure [in sulphate]? Did the other species also exhibit the same diurnal structure? What do you mean by "specific to each element"?

**Authors: We add the following sentence to be clearer: "*Each element exhibits its own type of daily variation in terms of amplitude and regularity*". Line 485-486**

**Reviewer 1:** Figure 7 This is a nice figure with LOTS of information contained within it. It's quite hard to imagine how these time series might look (the equivalent of Fig5+6). So again, this would support including the other species information in the supplementary information.

**Authors: We agree with this comment, which meets concerns raised by the reviewer 2 (see below). We think that the calculations performed and generalized on all elements measured are meaningful to the reader however we think that having the other detailed time series would simply be too much for such a paper. We remove the part of the discussion "Generalization" to the supplementary information for both part of the discussion "5.2 What is revealed by a higher sampling frequency?". Line 513-523 and in the part "5.3 What is revealed by better analytical precision?". Line 570-573. Please see supplementary information.**

**Reviewer 1:** Line 397-398 "artificially degraded the signals by adding a normally distributed noise" Nice idea!

Figure 8 Could you make these into line plots (i.e. connect your dots) so that we can see the noise chronologically? And also include the vertical lines (like in Figure 5) from the peak discharge down through the plots.

**Authors: We tried to add the vertical lines as eye guides but found that connecting all dots makes the figure unreadable.**

**Reviewer 1:** Figure 9 Again, make these into line plots? Also add a hydrograph.

**Authors: We found that connecting all dots makes the figure unreadable.**

**Reviewer 1:** Line 474 Is this a paraphrase or a direct quote?

**Authors: A direct quote. We add the reference to Kirchner et al. 2004 with the corresponding line number. Line 623**

**Reviewer 1:** Line 475-477 Nice punchline. But (and I apologize for being really pedantic here), the quote is to 'hear' the notes, not to play them. The stream is playing the notes. Your RL is therefore. . . a really sophisticated hearing aid. . .???! (Maybe a better simile needed).

**Authors: This comment is also formulated by the reviewer 2, we change the sentence word "play" by "hear" and the sentence: "*The improvements made possible by the RL allow us to hear the full potamological symphony.*" Line 624.**

**Reviewer 2:** General comments

**Reviewer 2:** This manuscript presents first data from a field deployment of an instrument package ("River Lab" or "RL") for high-frequency analysis of natural streamwaters. The instrumentation includes dual-channel ion chromatography and various physical and electrochemical probes. Data are presented to illustrate the quality of the chemical time series that can be obtained.

This manuscript makes a useful contribution to the growing literature on field-based chemical analyses of natural waters. In particular, the data presented here show that on-line analyses can be much more precise than those based on conventional sampling with later analysis in the lab.

However, the manuscript's characterization of the River Lab as a "breakthrough" (three times!) is not appropriate. Many other studies have also deployed wet chemistry instruments in the field (see Rode et al. ES&T, 50, 10297, 2016, and similar reviews). For example, a Swiss research group has recently published (also in HESS) a field deployment of another on-line ion chromatography system. That system arguably goes beyond the one presented here (it also includes isotopic analysis, and analyzes both rainfall and stream water, rather than just stream water alone), but the Swiss group does not use self-congratulatory terms like "breakthrough" to describe their work.

**Authors: We are of course aware of the studies conducted by the Swiss group. We found it very interesting. We have already mentioned in our manuscript Line 96. However one could state that our manuscript also "arguably goes beyond the" paper by the Swiss research group in terms of analytical performances and their interest for hydrochemistry**.

**Reviewer 2:** Likewise the claim that "... the high frequency and high precision of the RL enabled unprecedented observations of the fine structure in hydrochemical time series" is exaggerated. Similarly detailed results have been obtained before for other analytes using other field instrumentation, as well as by the Swiss group for major ions using IC. The present manuscript does a good job of demonstrating the precision of the RL, and does so in greater detail than I have seen before. But the observations themselves (at least the ones that are actually shown here) do not merit superlative terms like "unprecedented".

**Authors: We appreciate the general comments of reviewer 2. We agree that the vocabulary is somewhat exaggerated and we will tone it down. However we notice that the word "breakthrough" that reviewer 2 does not like is used by reviewer 1 in his/her review.**

**Reviewer 2:** Heroic claims like "Our study opens a new era of investigation into the fine structure of the hydrochemical signal in rivers" (line 478) will not create a favorable impression in the community of investigators who have already been working, in some cases for decades, on these questions. Similarly, claims like "Recording such fine stream hydrochemical variations is thus offering a new perspective on Critical Zone" (line 483) are unproven and should be removed, since no inferences about any catchment processes are derived in the present manuscript.

**Authors: We partly agree with the reviewer that the words need more attention but we really think that our study is providing new information on the fine structure of river chemistry over such a long period of time. It was not our intention to make the reader**

**feels that nothing was done before our study. We remove the most "heroic claims". we change breakthrough by "The ability". Line 109. We also remove the word "*unsuspected*". Line 117**

**Reviewer 2:** Several time series are shown to illustrate the results that can be obtained with the RL, and to illustrate the results that would be expected at lower sampling frequencies. This is fine as far as it goes, but given that the manuscript claims that the system has been measuring seven ions continuously for over a year, it seems strange that only a few days of data, for only two ions, are shown. Where is the rest of the data set, for the rest of the ions? It is understandable that the authors want to defer the analysis of the longer-term data set for a later paper, but they should at least demonstrate its existence by showing it to the audience.

**Authors: We understand the reviewer's comment but we would like to emphasize that in this article we first want to present the "proof of concept". This paper is thus not aimed at presenting the whole data acquired in one year. As correctly guessed by the reviewer, we "want to defer the analysis of the longer-term data set for a later paper". We only selected carefully two characteristic periods to illustrate the importance of recording river chemistry at high frequency and with a good precision. Presenting results over longer periods may dilute our conclusions. As stated in the introduction (and well understood by the reviewer 1) our paper aims at presenting the feasibility of measuring river chemistry by transporting the lab instruments to the field, not "giving away" a full year of measurements.**

**Reviewer 2:** To quantify the effects of subsampling and sampling error, the manuscript calculates how they affect the first four moments (mean, standard deviation, skewness, and kurtosis) of the distribution of measured concentrations for two brief sampling periods. Despite the time and space devoted to this analysis (two entire figures and parts of four others), it is not well posed and adds little to the paper, for the following reasons:

1) The moments of concentration distributions are rarely subjects of interest, particularly over such short periods of time.

**Authors: This is a totally unsupported statement that would deserve to be developed to warrant a proper answer. But we emphasize that we use these moments as a comparison tool between signals (original and altered) covering the *same periods of time*. In addition, we are doing this not in the scope of retrieving information on actual processes (which is probably what the reviewer has in mind when he/she says "subjects of interest"), but just in a sort of sensitivity analysis.**

**Reviewer 2:** 2) The distributions (particularly in Fig. 5) are sensitive to the interval of time that is considered. In particular, much comment is made about the "bimodal" distribution during the "flood event", but this is largely an artefact of the specific time interval that is chosen.

**Authors: We totally agree. However we carefully chose the interval between the very start and the very end of the flood event in order to not "under sample" this food event. By choosing a larger interval we would have altered the distribution by adding values from the non-flood period. Actually, reading our manuscript again we do not find that "much comment" is made about this bimodal distribution. We also note that many of the flood events we observed since the RL has been started show several peaks, such that choosing another flood event would not result in a different interpretation.**

**Reviewer 2:** 3) Skewness and kurtosis are mostly useful in characterizing uni-modal distributions, and their application to bimodal distributions is not very helpful.

**Authors: It all depends on what the reviewer means by "useful". It is true that for multi-modal distributions, the interpretation of skewness and kurtosis is not straightforward. We decide to remove the Kurtosis parameter to all of our study because it is not clear and do not give necessary information. However, we keep the skewness parameter but we present in the figure 5, 6, 7 and 8 and generalized to all elements in the supplementary information in the figure SI 5, 6, 7 and 8 with absolute value.**

**Reviewer 2:** 4) Statistical moments calculated from only five data points (as in Fig. 5) really should not be taken seriously!

**Authors: We agree with the reviewer, this is slightly misleading (in the sense that with such low sample numbers, the empirical estimators of the statistical moments have no chance to be close to their actual values), and we therefore removed the estimated statistical moments from the strongly sub-sampled signals (1/day) in Figs 5 and in the generalization Figure SI3.**

**Reviewer 2:** 5) The error bars in Figures 7 and 10 are unrealistic estimates of the uncertainties in the data points, because they do not account for the rather strong autocorrelation in the time series.

**Authors: We agree and we propose to remove these error bars, and to replace them by a "cloud of points" corresponding to each realization for the skewness calculation in the supplementary information. Please see figure SI 5, 6, 7 and 8.**

**Reviewer 2:** 6) Skewness and kurtosis are not ratio-scale variables (they do not have a natural zero), so calculating percentage changes in them makes no mathematical sense, for the same reason that a temperature increase from 1 to 21 degrees Celsius is not "an increase of 2,000 percent!"

**Authors: We understand the reviewer concern: unlike average and standard deviation, skewness and kurtosis are not ratio scale variables. We remove the Kurtosis parameter to all the paper and we present the absolute values of these metrics with a "cloud of points" corresponding to each realization for the skewness in the supplementary information figure SI 5, 6, 7 and 8.**

**Reviewer 2:** 7) The changes in distributions and statistical moments shown in Figures 8-10 are unsurprising to anyone with even a modest background in statistics, given that we are mixing the distributions of the original data with an assumed error distribution that has a mean, skewness, and kurtosis of zero, plus a standard deviation that is substantial compared with that of the original data.

**Authors: We are not sure to understand the reviewer's point. Results presented in Figs. 8-10 show that the mean is not affected by adding this normal noise, but all three other moments are. We believe that this will not be easily predicted by readers having "a modest background in statistics". And let alone quantitatively predicted - which is another interest of this approach: actual numbers are given. For example, our initial guess would have been, if anything, that only the standard deviation would be affected, since the standard deviation of the added signal is its only non-zero statistical moment.**

**Reviewer 2:** 8) Statements like "the average is not sensitive to analytical precision" (line 429) are self-evident (of course it isn't, as long as the added noise has a mean of zero!).

**Authors: This is true and our intention was not to present this as a surprise. We remove the sentence about the average. Line 571.**

**Reviewer 2:** The time series plots show the effects of subsampling and added noise very clearly. The statistical analysis does not help (and is often misleading), and should be omitted.

**Authors: As explained above, we do not agree with the reviewer about the interest of the statistical analysis (which allows us to build Figs. 7 and 10, and therefore to generalize the approach to all elements and report the results in little space, which would be impossible using the time series only). Therefore we will leave this statistical analysis.**

**Reviewer 2:** The manuscript is mostly well written, but in some places the grammar and word us- age need improvement. To take just the two examples that arise in the title itself: "progresses" is not a noun in English, and "Potamochemical" is not a word in any language. While one can appreciate the authors' creativity here, the use of a made-up word like "Potamochemical" will seem pedantic to many readers. A few of us may still recall that "potamology" means "study of rivers", but the usage of this term has been declining steadily for about 50 years and there is no compelling reason to revive it. Readers should not need to reach for their dictionaries to look up obscure Greek roots of words, just to understand the title of a scientific paper.

**Authors: We can understand the reviewer's feeling, which is exactly the opposite of that of reviewer 1. It is a personal point of view. On our side, we do not think that, because a term is declining, it should fade away. There is no intent in trying to look smart in this article, but we think that the almost forgotten term "potamology" should be revived so it could increase the visibility of our river and critical zone communities. We are not considering changing this word. However, we will "*progresses*" to "*progress*". Line 1**

**Reviewer 2:** Specific comments

The abstract claims that the RL was deployed "for over a year of continuous analyses" but the conclusion refers to measurements made "over several months". Which is it? Please provide plots showing "over a year of continuous analyses" of all ions, to substantiate the claim made in the abstract.

**Authors: We refer the reviewer to its own sentence above: we "want to defer the analysis of the longer-term data set for a later paper" in preparation. This is absolutely unnecessary for this "proof-of-concept" paper.**

**Reviewer 2:** The abstract says that the daily oscillations observed during the summer drought were "unexpected". Why were these oscillations unexpected, given that they have been reported in many previous papers, including several that are cited here?

**Authors: We remove the word "*unexpected*". Line 608**

**Reviewer 2:** The characterization of sampling frequency in millihertz (i.e., 40-minute sampling is described as 0.42 mHz) is unhelpful and will strike many readers as pedantry. If one wants to speak in terms of frequency, a more natural time base is 1 day (40-minute sampling is 36/day, 7-hour sampling is 3.42/day) or 1 week (40-minute sampling is 252/week, 7-hour sampling is 24/week).

**Authors: We change in the manuscript the frequencies with "40-minutes", "7-hourly" and "1/day". We remove all terms in "hertz" as we realize that is note clear for the readers.**

**Reviewer 2:** The "flood event" represents an increase in flow of only about 50% or so, and even the highest flow in figure 5 (about 200 L/s) is 50 times smaller than the reported peak flow of over 10 m3/s (see line 105). In what sense is it appropriate to call this a "flood event"?

**Authors: Indeed. To be fairer, we change the term "*flood event*" to "*rain event*".**

**Reviewer 2:** Figure 1 is artistic but less informative than it should be. Please provide a proper schematic of the instrumentation instead. Readers should be able to build a working version of the instrumentation (or at least understand the challenges involved in doing so) from the information provided.

**Authors: We think that this figure is a very good compromise between readability, such that the average reader will understand at a glance what the equipment is about, while the more interested reader will obtain first-order technical information from it. We therefore keep this figure in the main paper. However, we add in the next version a technical sketch with all information needed in the supplementary information Figure SI1.**

**Reviewer 2:** Figure 3 focuses the reader's attention on the conductivity data (which is not the subject here), because it is darker than the Cl data. Plot the conductivity data in a light color and the Cl data in a dark color, and connect the Cl points. And don't show the artificial 0-to-100 scale; show the real concentrations and conductivities.

**Authors: Absolute values for each measurement are already given in the figure. We still think that a normalized scale help the reader to focus on the relative delay to appreciate the cross-contamination and not the absolute value. However, we change the size of the dots for the Cl concentration to be clearer. Please see the figure 3.**

**Reviewer 2:** Figure 4 and associated text: why are cations shown for one date and anions shown for another date? Show all the ions for one date, so they can be compared. Also, the fonts in Figure 4 are inconsistent, and the placement of the panels is erratic. Didn't anyone notice this? And the two colors look the same in grayscale; again, didn't anyone check? Also, please show the real hour of the real day, not an artificial scale (I went crazy trying to figure out what was going on here, before I discovered that the scale was fake and the cations and anions have nothing to do with one another because they are on different days (but why does chloride stop before the other anions?).

**Authors: We apologize for these mistakes, and all of the requested format changes has been carried out. We also state more explicitly that these analyses were carried out over different days for anions and cations (which was necessary for logistical reasons, but which does not impair our approach). Please see the figure 4.**

**Reviewer 2:** The laboratory analyses were reportedly done on "IC devices similar to those installed in the RL", but what does "similar to" mean? Specify exactly what instruments those were, what columns were used, how old they were, and so on. This is important, because the relatively poor performance of the laboratory analyses is attributed to the difference in sampling, rather than the difference in instrumentation. If the laboratory instruments are an earlier generation of IC, or are using older columns, or have not been maintained as well, or

have not been operated as carefully, or, or, or... it is easy to see how the laboratory results could look poorer for many different reasons.

**Authors: As already answered to the reviewer 1, such information is already given in the supplement. However, we add in the manuscript:** *"…using IC devices similar to those installed in the RL (Thermo Fisher® ics 2100). In the laboratory, measurements were performed using Thermo Fisher® ics 5000 for cations measurements and Dionex® 120 from Thermo Fisher® for anions measurements. The calibration procedure in both laboratory and RL is the same using the same set of calibration solutions."* **Line 347-351.**

**Reviewer 2:** The obvious offset between the laboratory results and those from the RL are particularly concerning. What efforts have been undertaken to verify that these do not indicate an artifact in either the RL or the laboratory?

**Authors: We invite the reviewer 2 to read our reply formulated earlier to the first paragraph of the review 1.**

**Reviewer 2:** In Figure 8, add error to the whole time series, not just the arbitrarily defined "event" period. Details (a partial list) 50: Rozemeijer, not Rozemeiler 51: Chapman et al is 1997, not 1996 (and it would be appropriate to cite Neal et al., Hydrological Processes 2531, 2013, here). 292: discretely, not discreetly many figures: probability, not "probality" (didn't anybody notice this?) 476: "the best orchestra available": clever phrase, but an unproven assertion and will be perceived as inappropriate bragging 477: "to play the potamological symphony": no, instruments *record* the symphony; nature plays it.

**Authors: We add these new references and correct the typos. We also invite the reviewer 2 to see our reply to reviewer 1 regarding the mistake between "*play*" and *hear*".**

---

## Referee Report (RR1)

Second review :

This is an improvement over the original submission, but there are still problems that need to be dealt with.  The general issue is that the manuscript still claims a level of originality and significance that is not justified by the evidence that is presented.  I assume that the authors are not intentionally shading the facts, but unfortunately this is the impression that the manuscript gives.

For example, the introduction is still written as if nothing like the authors' River Lab has ever been built before.  But various "field lab" setups have been built, many times.  Indeed, there was a paper published in HESS earlier this year describing a field lab based on ion chromatography that is very similar to the River Lab.  The authors are well aware of this prior work (and indeed it was pointed out in a previous review), but they still refuse to acknowledge it.  Instead, for example, they only cite the HESS paper following a statement about "issues related to sample transport, filtration and storage".  Refusing to acknowledge prior work and instead citing it for trivial or tangential points is inappropriate and one would hope that the authors would recognize this.

The manuscript says that "online instrumental devices in which continuously pumped water is injected have been suggested as an alternative to monitor water chemistry."  Such systems have not only been "suggested", they have actually been built and used, in some cases for many years.

The manuscript continues, "To date, these systems have only been used to monitor nutrients such as dissolved N or P."  This is false and the authors know that it is false.  They know perfectly well that another group has already published an ion chromatography system very similar to theirs and that measures the same ions that they have measured, but their manuscript appears to have been carefully written to conceal that fact.

Likewise the authors have not acknowledged the major recent overview of high-frequency sampling applications (Rode et al., 2016), even after this was pointed out to them in the previous round of review.

The slanted presentation extends to technical matters as well.  The manuscript simulates the addition of 2% and 4% noise, saying that these are "representative of the relative analytical precision reported for most laboratory IC devices (Neal et al. 2011; Aubert et al., 2013a)."  The implication is that the data from Neal et al. or Aubert et al. would look as messy as the simulations presented here, but this is a gross distortion.  For example, the precisions of the IC measurements of SO4 in Neal et al. are about 2%, but at concentrations of only 2 ppm, rather than the roughly 60 ppm presented here.  Because IC noise percentages are typically higher at lower concentrations, the Neal et al. measurements would likely not be much worse (and possibly better) than the RL measurements at comparable concentrations.

The data from the River Lab look very nice, and it is neither necessary nor appropriate to try to make them look better by misrepresenting the accomplishments of others.

The comparison of the River Lab data and the IGPG lab data depends critically on the calibrations that are used for the two instruments.  Here the manuscript directly contradicts itself.  On line 268 the reader is told, "The calibration procedure in both laboratory and RL is the same using the same set of calibration solutions."  But just one page later, in explaining the different results obtained from the two instruments, the manuscript says, " In addition, the most accurate measurements were obtained with the RL rather than with the laboratory equipment because (1) the calibration curve of

the RL was made from a series of solutions (dilutions of the "River x1" solution) having the same element ratios as the solution used for the accuracy test (the "River x1" solution) ... with our in-lab IC instruments ...we used a series of calibration solutions having the same concentration for all elements..."  This is a rather obvious discrepancy and it is surprising that apparently none of the authors have noticed it, even after the issue of calibrations was raised in the previous review.

There are language problems here that should not be present in submission to a major international journal (particularly after revision).  The SI is particularly bad; after finding 14 language errors in just two pages, I stopped counting.  There are 10 authors on this paper, some of whom are really good at scientific English, and if they have really all read and approved the manuscript it is hard to understand how so many errors could still persist.  Apparently nobody has even run a spell-checker; otherwise bloopers like "ratther" would have been caught.

---

## Author Response (AR2)

**Part 1. (Paul Floury on behalf of the co-authors).**

Please find a new version of our manuscript "The Potamochemical symphony: new progress in the high-frequency acquisition of stream chemical data" with the corresponding associated content.

We are really sorry about the tone the review has taken and we did our best in the revised version to take the (positive) comments and suggestion of the reviewer into account. We present a new version of the paper which has been significantly reworked. We understood that some sentences of the initial manuscript were hurtful for the reviewer and did not give enough credit to his work. We apologized if sentences give him the feeling that we deliberately wanted to refuse acknowledge previous studies or even worst, disregard them. The fact is that the design of the River Lab that we describe in this paper and the design of the lab-in-the-field Swiss hut were contemporary. The Swiss paper (von Freyberg et al., 2017) was published first, what we totally humbly acknowledge now in the paper. We have changed the introduction to make it clear that the River Lab is a parallel initiative to the Swiss hut. Our intention was not to hurt.

In the following, we take the point by point list of comments raised by the reviewer and specify the corresponding changes made to the ms. All reviewer's comments has been taken into account and accepted.

> **Reviewer:** This is an improvement over the original submission, but there are still problems that need to be dealt with. The general issue is that the manuscript still claims a level of originality and significance that is not justified by the evidence that is presented. I assume that the authors are not intentionally shading the facts, but unfortunately this is the impression that the manuscript gives.

We apologize if some sentences in the manuscript gave the feeling to misrepresent previous studies in the field. This was not our goal. We reformulated all sentences mentioned by the reviewer (see in the following).

> **Reviewer:** For example, the introduction is still written as if nothing like the authors' River Lab has ever been built before. But various "field lab" setups have been built, many times. Indeed, there was a paper published in HESS earlier this year describing a field lab based on ion chromatography that is very similar to the River Lab. The authors are well aware of this prior work (and indeed it was pointed out in a previous review), but they still refuse to acknowledge it. Instead, for example, they only cite the HESS paper following a statement about "issues related to sample transport, filtration and storage". Refusing to acknowledge prior work and instead citing it for trivial or tangential points is inappropriate and one would hope that the authors would recognize this.

**The introduction was reworked. We now present the achievements of the Swiss group the reviewer is referring too as follow:**

"A new solution for high1frequency measurement of river chemistry is offered by bringing the laboratory's measuring devices to the field (the "lab in the field" concept). A Swiss group has recently successfully developed such a system (von Freyberg et al., 2017) by installing ionic chromatography devices in a hut next to a stream. In this paper, we present a parallel initiative named as the River Lab (RL) and funded by French program CRITEX: "Innovative sensors for the temporal and spatial EXploration of the CRITical Zone at the catchment scale" (https://www.critex.fr)".

**Line 76 was confusing and therefore has been changed. It now reads as:**

"This approach, like the previously published one, overcomes traditional limitations on the number of samples and avoids several issues related to sample transport, filtration and storage"

> **Reviewer:** The manuscript says that "online instrumental devices in which continuously pumped water is injected have been suggested as an alternative to monitor water chemistry." Such systems have not only been "suggested", they have actually been built and used, in some cases for many years.

We reformulate correctly this part of the introduction. The sentence in question now read as:

"Several papers have been published over the last decade reporting existing devices mostly focused on monitoring dissolved N or P and organic matter (Kunz et al., 2012; Clough et al., 2007; Aubert et al., 2013a; Aubert et al., 2013b; Rode et al. 2016)"

> **Reviewer:** The manuscript continues, "To date, these systems have only been used to monitor nutrients such as dissolved N or P." This is false and the authors know that it is false. They know perfectly well that another group has already published an ion chromatography system very similar to theirs and that measures the same ions that they have measured, but their manuscript appears to have been carefully written to conceal that fact.

We have rewritten the introduction that we now think is overcoming the issue raised by the reviewer. We separated the devices allowing people to measure nutrients and organic matter at high frequency, usually based on is situ sensors from the "lab in the field" concept now represented by the Swiss and our group initiatives.

We also quoted in the conclusion von Freyberg et al. (2017) to acknowledge previous work and emphasizing the fact that both studies (our and von Freyberg et al., 2017) were conducted simultaneously:

" The improvements made possible by the RL here or concomitantly by von Freyberg et

al. (2017) allow us to consider hearing the full potamological symphony"

**Reviewer:** Likewise the authors have not acknowledged the major recent overview of high-frequency sampling applications (Rode et al., 2016), even after this was pointed out to them in the previous round of review.

Authors: We have added this reference in the new version of the introduction and give credit to this publication offering an overview of in situ commercial probes.

"A recent overview of the potential of available conductivity, dissolved oxygen and carbon

dioxide, nutrients, dissolved organic matter, chrlorophyll and Co in situ probes is given by

Rode et al. (2016)"

**Reviewer:** The slanted presentation extends to technical matters as well. The manuscript simulates the addition of 2% and 4% noise, saying that these are "representative of the relative analytical precision reported for most laboratory IC devices (Neal et al. 2011; Aubert et al., 2013a)." The implication is that the data from Neal et al. or Aubert et al. would look as messy as the simulations presented here, but this is a gross distortion.

The goal of the discussion was not to disregard the previous work but just to highlight an improvement in the precision. In order to avoid any misinterpretation, we deleted the two references in Line 553 "(Neal et al. 2011; Aubert et al., 2013a)."

"Noise levels of 4% and 2% were tested as they are representative of the "standard' analytical

precision reported for most laboratory IC devices"

**Reviewer:** For example, the precisions of the IC measurements of SO4 in Neal et al. are about 2%, but at concentrations of only 2 ppm, rather than the roughly 60 ppm presented here. Because IC analytical noise expressed as a percentage typically decreases as

concentrations go up, the Neal et al. measurements would likely not be much worse (and possibly better) than the RL measurements at comparable concentrations. The data from the River Lab look very nice, and it is neither necessary nor appropriate to try to make them look better by misrepresenting the accomplishments of others.

We apologize if the manuscript gives the impression to the reviewer that we misrepresent previous works. This was an awkward way of writing as the only goal of the exercise proposed in the discussion was to highlight the added-value of the RL permitted by the improvement of precision. We hope the new formulation will satisfy the reviewer and clears up misunderstanding.

**Reviewer:** The comparison of the River Lab data and the IGPG lab data depends critically on the calibrations that are used for the two instruments. Here the manuscript directly contradicts itself. On line 268 the reader is told, "The calibration procedure in both laboratory and RL is the same using the same set of calibration solutions." But just one page later, in explaining the different results obtained from the two instruments, the manuscript says, " In addition, the most accurate measurements were obtained with the RL rather than with the laboratory equipment because (1) the calibration curve of the RL was made from a series of solutions (dilutions of the "River x1" solution) having the same element ratios as the solution used for the accuracy test (the "River x1" solution) ... with our in-lab IC instruments ...we used a series of calibration solutions having the same concentration for all elements..."
This is a rather obvious discrepancy and it is surprising that apparently none of the authors have noticed it, even after the issue of calibrations was raised in the previous review.

Authors: This is a mistake remaining from the first version. We deleted the following sentence Line 331: "(1) the calibration curve of the RL was made from a series of solutions (dilutions of the "River x1" solution) having the same element ratios as the solution used for the accuracy test (the "River x1" solution)". We are sorry about this confusion.

There are language problems here that should not be present in submission to a major international journal (particularly after revision). The SI is particularly bad; after finding 14 language errors in just two pages, I stopped counting. There are 10 authors on this paper, some of whom are really good at scientific English, and if they have really all read and approved the manuscript it is hard to understand how so many errors could still persist. Apparently nobody has even run a spell-checker; otherwise bloopers like "ratther" would have been caught.

Authors: We modified "ratther" to "rather". We also checked all language mistakes in the supplementary content. We are sorry about this negligence.

Other additions.

- We added in Fig. 1 a photography of the River Lab.
- We correct the figure caption in the manuscript
- We add a picture of the RiverLab in the figure 1.

- We added a reference of high-frequency measurement in rivers in the introduction (Escoffier, N., Bensoussan, N., Vilmin, L., Flipo, N., Rocher, V., David, A., ... & Groleau, A. (2016). Estimating ecosystem metabolism from continuous multi-sensor measurements in the Seine River. *Environmental Science and Pollution Research*, 1-17.).